A new metriacanthosaurid theropod dinosaur from the Middle Jurassic of Yunnan Province, China

Zou Yi 1 2
Chen Li 3
Wang Tao 4
Wang Guo-Fu 3
Zhang Wei-Gang 5
Zhang Xiao-Qin 6
Wang Zhen-Ji 6
Wu Xiao-Chun 7
You Hai-Lu youhailu@ivpp.ac.cn 1 2
1 Key Laboratory of Vertebrate Evolution and Human Origins, Institute of Vertebrate Paleontology and Paleoanthropology, Chinese Academy of Sciences , Beijing , China
2 College of Earth and Planetary Sciences, University of Chinese Academy of Sciences , Beijing , China
3 Chuxiong Prefectural Museum , Chuxiong , China
4 Center for Dinosaur Research and Protection, Bureau of Natural Resources of Lufeng City , Lufeng , China
5 Chuxiong Jurassic Cultural Tourism Industrial Park Development Co. Ltd , Chuxiong , China
6 Chuxiong Normal University , Chuxiong , China
7 Canadian Museum of Nature , Ottawa , Canada
Johnson Michela
Electronic publication date: 2025 Apr 2
Publication date: 2025
Volume: 13
Electronic Location ID: e19218
Received 2024 Sep 26; Accepted 2025 Mar 6
Copyright: ©2025 Zou et al.
Copyright year: 2025
Copyright holder: Zou et al.
License: This is an open access article distributed under the terms of the Creative Commons Attribution License, which permits unrestricted use, distribution, reproduction and adaptation in any medium and for any purpose provided that it is properly attributed. For attribution, the original author(s), title, publication source (PeerJ) and either DOI or URL of the article must be cited.
License URL: https://creativecommons.org/licenses/by/4.0/

Keywords: Metriacanthosauridae, Theropod, Middle Jurassic, Yunnan Province

Funding: National Natural Science Foundation of China 42288201 42372030 42002014 Beijing Natural Science Foundation 5224037 Expert Local Level Scientific Workstation of Yunnan Province This work was supported by the National Natural Science Foundation of China (42288201, 42372030, and 42002014), the Beijing Natural Science Foundation (5224037) and Expert Local Level Scientific Workstation of Yunnan Province. The funders had no role in study design, data collection and analysis, decision to publish, or preparation of the manuscript.

==============================
Metriacanthosaurid theropods represent a basal-branching lineage of tetanurans. Members of this clade are mainly medium to large-sized and lived in Laurasia during the Middle Jurassic to the Early Cretaceous. In this clade, Sinraptor dongi, Sinraptor hepingensis, and Yangchuanosarus shangyouensis from the Late Jurassic are well represented by the nearly complete specimens, but the incompleteness of Middle Jurassic taxa hinders our knowledge of the origin and early evolution of Metriacanthosauridae. This paper describes a new genus and species of metriacanthosaurids, Yuanmouraptor jinshajiangensis gen. et sp. nov, from the Middle Jurassic Zhanghe Formation of Yunnan Province, China. The new taxon is represented by a cranium and the anterior section of the vertebral column including the complete cervical series and the first dorsal vertebra. Yuanmouraptor jinshajiangensis can be diagnosed based on the following autapomorphies: the anterior process of postorbital sheet-shaped and keeping consistent depth; ventral ramus of postorbital bearing a laterally twisted trough running along its lateral surface; ventral surface of axial intercentrum parallel with that of axial centrum; discontinuity of inclination on anterodorsal margin of the third and fourth cervical vertebrae; strongly posteriorly elongated epipophyses of anterior cervical vertebrae; deeply excavated pneumatic foramina on the third cervical vertebra; sheet-shaped and subrectangular neural spines of posterior cervical vertebrae. Phylogenetic analysis recovers Yuanmouraptor as the most basal-branching member within Metriacanthosauridae and provides a new alternative phylogenetic topology of non-coelurosaurian tetanurans.

Introduction

Metriacanthosauridae is a family of medium-to-large sized carnivorous dinosaurs and represents a basal-branching clade within the Allosauroidea (Holtz, Molnar & Currie, 2004; Smith et al., 2007; Benson, 2010; Carrano, Benson & Sampson, 2012; Hendrickx, Hartman & Mateus, 2015; Coria & Currie, 2016; Rauhut, Hübner & Lanser, 2016; Rauhut & Pol, 2019; Rauhut et al., 2024; Lamanna et al., 2020). Some researches claim that metriacanthosaurids possess closer relationship with carcharodontosaurids (Coria & Currie, 2002; Allain, 2002; Rauhut, 2003; Kellermann, Cuesta & Rauhut, 2025), rendering Metriacanthosauridae a more derived group within Allosauroidea. No matter what position Metriacanthosauridae has within Allosauroidea, members of this clade mainly came from the Middle to Late Jurassic strata of western China (Fig. 1), such as Sichuan, Chongqing, Xinjiang, and Yunnan (Dong et al., 1978; Dong, Zhou & Zhang, 1983; Gao, 1992; Gao, 1993; Gao, 1999; Currie & Zhao, 1993; Wu et al., 2009). Apart from those taxa found in China, metriacanthosaurid theropods were also reported in the Late Jurassic of England (Huene, 1923; Walker, 1964), the Late Jurassic of Kyrgyzstan (Rauhut et al., 2024), and the Late Jurassic and Early Cretaceous of Thailand (Buffetaut, Suteethorn & Tong, 1996; Samathi, Chanthasit & Sander, 2019). Recently, Yu et al. (2023) reported a probable distribution of this clade in the Tibetan Plateau. Here we report a new genus and species of Metriacanthosauridae collected from the Middle Jurassic Zhanghe Formation of Jiangyi Township, Yuanmou County of Yunnan Province, China (Fig. 1). Our phylogenetic analysis suggests that the new taxon is probably one of the two most basal-branching metriacanthosaurids. Furthermore, some characters presented in the new taxon are also shared with several megalosauroids (Li et al., 2009; Dai et al., 2020) and non-tetanuran theropods (Colbert, 1989; Carrano, Loewen & Sertich, 2011; Marsh & Rowe, 2020), which suggests that these shared characters were gained independently by the aforementioned taxa.

Figure 1 Geographical distribution of metriacanthosaurid theropods in Yunnan, Sichuan, and Chongqing, China.

Each number indicates an individual: 1, Shidaisaurus jinae; 2, ‘Szechuanosaurus’ zigongensis; 3, CNM V214; 4, Sinraptor hepingensis; 5 & 6, Yangchuanosaurus shangyouensis.

Geological setting

Central Yunnan region received continental deposit during the Jurassic, and formed a suite of thick red siltstone bed. It was divided into Chuxiong and Kunming subregions on the west and east sides respectively (Fang et al., 2008; Cheng et al., 2004). Zhanghe Formation is representative Middle Jurassic deposit of Chuxiong subregion in central Yunnan region, and it was first established by Yunnan regional survey team in 1961 (Zhang et al., 1996) or 1965 (Bureau of Geology and Mineral Resources of Yunnan Province, 1990) at Zhanghe Village, Xiangyun County of Yunnan. Due to the continental sedimentary nature of Zhanghe Formation, the absolute age of this unit remains unclear. But its lithologic and biostratigraphic characteristics are comparable to those of once-called Upper Lufeng Formation, which was reassigned as Middle Jurassic Chuanjie Formation and Laoluocun Formation from bottom to top (Bureau of Geology and Mineral Resources of Yunnan Province, 1990; Zhang et al., 1996; Cheng et al., 2004). Zhanghe Formation in the research area is generally a set of red argillaceous siltstone, with red sandstone interbedded in it. West to the fossil locality, Zhanghe Formation conformably overlaps the top of the Early Jurassic Fengjiahe Formation, which is mainly a set of yellow thick sandstone. Before this new finding of theropod, Zhanghe Formation has yielded sauropods including Yuanmousaurus (Lü et al., 2006) (Fig. 2), Eomamenchisaurus (Lü et al., 2008), and Nebulasaurus (Xing et al., 2013). A probable sauropod ‘Shunosaurus’ jiangyiensis (Fu & Zhang, 2004) was also found in Zhanghe Formation, but its validity needs further examination (Ma et al., 2021). The underlying Fengjiahe Formation also has yielded the sauropodomorph Yunnanosaurus youngi (Lü et al., 2007; Ren et al., 2021) (Fig. 2).

Figure 2 Geographic location of the fossil localities around the Jiangyi Township and the general geological map of this area.

All silhouettes of dinosaurs are drawn by Yi Zou.

Material & Methods

The new specimen LFGT-ZLJ0115 studied here was excavated by a field team led by Guo-Fu Wang and De-Zhi Liu of Chuxiong Prefectural Museum in March 2006. The specimen was unearthed from a layer of thick and hard red siltstone (Fig. 3) of the Middle Jurassic Zhanghe Formation, and the locality is surrounded by farmland now. The specimen is now on display in the museum of Lufeng World Dinosaur Valley in Lufeng City, Yunnan Province. The specimen includes a relatively complete skull and the first 11 vertebrae including 10 cervical vertebrae and the anterior-most dorsal vertebra. Most cranial bones are still in articulation or closely associated. Some of the cranial elements are heavily distorted or covered by matrix or other bones, rendering difficulty in determination of bone sutures or internal structures. The specimen was prepared using mechanical tools (pneumatic chisels) and photographed from various perspectives with a Sony DLSR-A700 digital camera. Line drawings were made based on the reference photographs and checked against the original specimens.

Figure 3 The locality of LFGT-ZLJ0115.

The specific layer yielding the specimens (A) and the surrounding environment (B).

Phylogenetic analysis

The new matrix for the phylogenetic analysis in this study was modified based on that of Carrano, Benson & Sampson (2012), which mainly focused on the phylogenetic relationship within tetanurans. We added five new and 26 characters modified from the datasets of Lamanna et al. (2020), Eddy & Clarke (2011), Brusatte & Sereno (2008), and Schade et al. (2023) (see the online File S1 for details). We added Panguraptor (You et al., 2014), Zuolong (Choiniere et al., 2010), Guanlong (Xu et al., 2006), and Eoabelisaurus (Pol & Rauhut, 2012) to the matrix to enrich the samples of basal neotheropod, Coelurosauria, and Ceratosauria, respectively. Several basal-branching tetanurans such as Asfaltovenator (Rauhut & Pol, 2019), Wiehenvenator (Rauhut, Hübner & Lanser, 2016), and Yunyangosaurus (Dai et al., 2020) were added because these taxa were recently reported tetanurans. Alpkarakush kyrgyzicus (Rauhut et al., 2024) (the most recently named Central Asian metriacanthosaurid) was also added to the dataset. The new matrix, consisting of 372 characters and 70 operational taxonomic units (OTUs), was analyzed using TNT v. 1.6 (Goloboff & Morales, 2023) with both equal weights and implied weights of characters. The most parsimonious trees (MPTs) were recovered by a traditional search of 1,000 replicates of Wagner trees followed by tree bisection and reconnection, with 10 trees saved per replication. In the implied weighting phylogenetic analysis, we set the k value equals to 3, 6, 9, 12 (referring to Rauhut et al., 2024 and Goloboff, Torres & Arias, 2018) to test the effect of homoplasy upon the phylogenetic result. None of the characters were treated as ordered.

Nomenclatural acts

The electronic version of this article in Portable Document Format (PDF) will represent a published work according to the International Commission on Zoological Nomenclature (ICZN), and hence the new names contained in the electronic version are effectively published under that Code from the electronic edition alone. This published work and the nomenclatural acts it contains have been registered in ZooBank, the online registration system for the ICZN. The ZooBank LSIDs (Life Science Identifiers) can be resolved and the associated information viewed through any standard web browser by appending the LSID to the prefix http://zoobank.org/. The LSID for this publication is: urn:lsid:zoobank.org:pub:2A9F32AD-B671-4F48-8A6E-0A69976A75FB. The online version of this work is archived and available from the following digital repositories: PeerJ, PubMed Central SCIE and CLOCKSS. The LSID for Yuanmouraptor jinshajiangensis is: urn:lsid:zoobank.org:act:5AE0D7CB-C337-41A2-BDC8-1F2E500624F6.

Results

Systematic paleontology

Dinosauria Owen, 1842	
Theropoda Marsh, 1881	
Tetanurae Gauthier, 1986	
Allosauroidea Currie & Zhao, 1993	
Metriacanthosauridae Paul, 1988	
Yuanmouraptor gen. nov.	

urn:lsid:zoobank.org:act:8F99DC0B-5E55-42CD-A0CF-9216F9EBE268

Diagnosis—As for the only species.

Yuanmouraptor jinshajiangensis gen. et sp. nov.

urn:lsid:zoobank.org:act:5AE0D7CB-C337-41A2-BDC8-1F2E500624F6

Etymology—The genus name, ‘Yuanmou’, refers to Yuanmou County where the holotype was collected, and ‘raptor’ is Latin for the robber. The specific name, ‘jinshajiang’ (namely the Jinsha River, the middle region of the Yangtze River) which passes through Yuanmou County and the type locality is located on the north bank of the river.

Holotype—LFGT-ZLJ0115: a partial skeleton consists of a nearly complete skull with mandible and 11 articulated anterior vertebrae including 10 cervical vertebrae and the first dorsal vertebra.

Type Locality and horizon—Xiabanjing Village, Jiangyi Township, Yuanmou County, Chuxiong Yi Autonomous Prefecture, Yunnan Province, China; Zhanghe Formation, early Middle Jurassic, Aalenian/Bajocian (Bureau of Geology and Mineral Resources of Yunnan Province, 1990).

Diagnosis—A medium-sized metriacanthosaurid dinosaur differing from other metriacanthosaurids by the following unique combination of characters (autapomorphies are indicated with an asterisk): an accessory foramen located within antorbital fossa on lacrimal and ventral to pneumatic foramen, similar to Allosaurus; dorsal part of lacrimal bearing a low rugosity, similar to megalosaurids; lack of pneumatic fenestra on lateral surface of jugal, shared with non-tetanurans; the anterior process of postorbital sheet-shaped and its depth keeping consistent*; ventral ramus of postorbital bearing a laterally twisted trough running along its lateral surface*; ventral surface of axial intercentrum parallel with that of axial centrum, shared with Piatnitzkysaurus and non-tetanurans; discontinuity of inclination on anterodorsal margin of the third and fourth cervical vertebrae, similar to that of Dilophosaurus and Baryonyx; flattened peripheral band on anterior articular surface of anterior cervical centra, shared with megalosaurids and some ceratosaurians; strongly posteriorly elongated epipophyses on anterior cervical vertebrae*; strongly ventromedially excavated pneumatic foramen on the third cervical vertebra*; sheet-shaped and subrectangular neural spines of posterior cervical vertebrae*.

General description of the cranium

The conditions of preservation of LFGT-ZLJ0115 are different between each side, and bones show many fractures which might be caused during preservation. On the left side (Figs. 4A, 4B), most parts of the nasal and elements around the orbit and lateral temporal fenestra are missing. On the right side (Figs. 4C, 4D), although the nasal is also poorly preserved, other bones are relatively more complete than those of the left side. The mandibular ramus is well preserved on both sides. The preserved elements of the skull and mandible are generally articulated, and thus most of the internal structures are obscured from observation except the right ramus of the mandible. The main fenestrae of the skull, such as the naris, antorbital fenestra, orbit, lateral temporal fenestra, and supratemporal fenestra, are all damaged or largely distorted. The preserved skull is measured 53.9 cm in anteroposterior length, and the reconstruction (Fig. 5) of the skull measures 60.1 cm in anteroposterior length. In comparison, the type specimen of Yangchuanosaurus shangyouensis (Dong et al., 1978) bears a skull length of 78 cm, and the referred specimen (Y. magnus, reported by Dong, Zhou & Zhang (1983), was considered to present different ontogenetic stage of Y. shangyouensis by Carrano, Benson & Sampson (2012)) has an estimated skull length of 111 cm. The skull of Sinraptor dongi (Currie & Zhao, 1993) is 90 cm long and the skull of S. hepingensis (Gao, 1992; Gao, 1999; here follows the assignment in Currie & Zhao, 1993; Carrano, Benson & Sampson, 2012) is 104 cm long.

Figure 4 Cranium of Yuanmouraptor jinshajiangensis gen. et sp. nov. (LFGT-ZLJ0115).

Cranium in (A) left lateral view with (B) labeled drawing and (C) right lateral view with (D) labeled drawing. Abbreviations: an, angular; ar, articular; bs, basisphenoid; d, dentary; f, frontal; j, jugal; l, lacrimal; lsp, laterosphenoid; m, maxilla; n, nasal; ot, otoccipital; prm, premaxilla; pa, parietal; pal, palatine; par, prearticular; po, postorbital; pop, paroccipital process; pr, prootic; prf, prefrontal; q, quadrate; qj, quadratojugal; sa, surangular; sp, splenial; so, supraoccipital; sq, squamosal. Striated area indicates damage and grey area indicates matrix. Scale bar represents 100 mm. Photos by Xiao-Chun Wu.

Figure 5 Reconstruction of the cranium of Yuanmouraptor jinshajiangensis gen. et sp. nov. (LFGT-ZLJ0115).

Abbreviations: an, angular; bs, basisphenoid; d, dentary; emf, external mandibular fenestra; j, jugal; l, lacrimal; m, maxilla; ot, otoccipital; prm, premaxilla; par, prearticular; pal, palatine; po, postorbital; pop, paroccipital process; pr, prootic; prf, prefrontal; q, quadrate; qj, quadratojugal; sa, surangular; sq, squamosal. Shaded area indicates the missing part, and dashed line marks the margin of breakage of bone. Scale bar represents 100 mm.

Premaxilla—Only the left premaxilla (Figs. 6A, 6B) is preserved, with most of the supranarial process missing except for its risen base. In lateral view, the premaxillary body (below the external naris) is roughly quadrangular and slightly higher than long (5.65 × 5.42 cm), which is similar to the condition of Ceratosaurus (Madsen & Welles, 2000), Torvosaurus (Britt, 1991), Majungasaurus (Sampson & Witmer, 2007), and Acrocanthosaurus (Eddy & Clarke, 2011). This is in contrast to Sinraptor (Currie & Zhao, 1993), Allosaurus (Britt, 1991), Neovenator (Brusatte, Benson & Hutt, 2008), Dubreuillosaurus (Allain, 2002), Marshosaurus (Madsen, 1976b), and Monolophosaurus (Brusatte et al., 2010a), in which the premaxilla is slightly longer than high. The ventral border of the external naris is nearly parallel with the premaxillary alveolar margin in Yuanmouraptor. This differs from the offset alveolar margin in many basal neotheropods such as Coelophysis (Colbert, 1989) and Dilophosaurus (Marsh & Rowe, 2020). The premaxilla of Yuanmouraptor bears four alveoli, which is a primitive condition for theropods (Allain, 2002; Sampson & Witmer, 2007, Currie & Zhao, 1993), as in Sinraptor, Yangchuanosaurus (Dong et al., 1978; Dong, Zhou & Zhang, 1983), but five alveoli are presented in Allosaurus (Madsen, 1976a) and Neovenator (Brusatte, Benson & Hutt, 2008). The fourth tooth is broken with only a little part of it preserved. The other three are complete and compressed labiolingually with slightly backward curvature. The distal carina is well developed and extended throughout the whole length, whereas the mesial carina is only visible at the epical one third of the second tooth in lateral view (Fig. 6C).

Figure 6 Premaxilla and maxilla of Yuanmouraptorjinshajiangensis gen. et sp. nov. (LFGT-ZLJ0115).

Left premaxilla in (A) lateral view with (B) labeled drawing. (C) Serration on the premaxillary teeth, with mesial carina pointed by white arrows. (D) Serration on the mesial and distal carina of the maxillary teeth. Left maxilla in (E) lateral view with (F) labeled drawing. Right maxilla in (G) lateral view with (H) labeled drawing. Abbreviations: aof, antorbital fossa; aofe, antorbital fenestra; asr, ascending ramus of maxilla; en, external naris; mc, maxillary contact; mf, maxillary fenestra; m1-14, maxillary teeth 1–14; nf, narial fossa; np, nasal process; pmf, promaxillary fenestra; prc, premaxillary contact; p1-3, premaxillary teeth 1–3; sn, subnarial process. Striated area indicates damage. Scale bars for (A–B) represent 50 mm, for (C–D), five mm, and for (E–H), 100 mm. Photos by Xiao-Chun Wu and Yi Zou.

The anterodorsal border of the premaxillary body is missing, along with the tip of the supranarial process (nasal process), but the preserved main part of the process is posterodorsally oriented and forms the anteroventral margin of the external naris. The narial fossa is located ventrally to the preserved part of the external naris as in Sinraptor (Currie & Zhao, 1993), Allosaurus (Madsen, 1976a), and Dubreuillosaurus (Allain, 2002). But in Acrocanthosaurus (Eddy & Clarke, 2011), the narial fossa is positioned further anteriorly. The anterior rim of the premaxilla is damaged, but based on the position of the first alveolus and mediolaterally constricted anterior margin, this part is slightly damaged. This suggests that the anterior margin of the premaxillary body might be nearly vertical as in Sinraptor (Currie & Zhao, 1993), Ceratosaurus (Madsen & Welles, 2000), and Allosaurus (Madsen, 1976a). Whereas in that of Torvosaurus (Britt, 1991), Dubreuillosaurus (Allain, 2002), and Duriavenator (Benson, 2008), the angle between the anterior margin of the main body of the premaxilla and alveolar margin is more rounded and the anterior margin of the main body is more posterodorsally inclined. The subnarial process (maxillary process) is relatively complete and of triangular-shape oriented posterodorsally in lateral view, resembling that of Ceratosaurus (Madsen & Welles, 2000), Neovenator (Brusatte, Benson & Hutt, 2008), and Sinraptor (Currie & Zhao, 1993) in relative size and orientation. The subnarial process in Acrocanthosaurus (Eddy & Clarke, 2011) and Allosaurus (Madsen, 1976a) is elongated dorsoposteriorly, whereas in Duriavenator (Benson, 2008) this process is more posteriorly oriented. The posteroventral rim of the subnarial process is confluent with the posterior border of the premaxillary body, and both form the slightly posterodorsally inclined suture with the maxilla in lateral view. The posterior border of the premaxillary body ventral to the subnarial process presents a rugose surface and indicates the contact with the maxilla (Fig. 6B). Whether Yuanmouraptor bears a subnarial foramen is not certain due to the blocking of matrix, but this structure is well developed in Allosaurus (Madsen, 1976a), Acrocanthosaurus (Currie & Carpenter, 2000), Sinraptor (Currie & Zhao, 1993), and most tentanurans, thus the subnarial foramen could also occur in Yuanmouraptor. Based on the suture of the premaxilla with the maxilla, there is no subnarial gap between these two bones, and the tooth row of each bone is continuous and at the same level. This differs from those in basal neotheropods (Colbert, 1989; Rowe, 1989; Marsh & Rowe, 2020) and spinosaurids (Sereno et al., 1998; Barker et al., 2021), which bear pronounced subnarial gap between premaxilla and maxilla.

Numerous foramina are mainly scattered on the lateral surface ventral to the mid height of the premaxillary body and open ventrolaterally, similar to the distribution pattern in Sinraptor and Yangchuanosaurus, whereas in many megalosaurids such as Dubreuillosaurus (Allain, 2002), Torvosaurus (Britt, 1991), and Marshosaurus (Madsen, 1976b) the foramina are mainly distributed on the anterior half of the premaxillary body. In Neovenator (Brusatte, Benson & Hutt, 2008), the foramina spread evenly over the lateral surface of the premaxillary body.

Maxilla—Both the left and right maxillae are adhered to the matrix, and thus the medial surface is obscured. The main body of the left maxilla (Figs. 6E, 6F) is well preserved but lacks most of its ascending ramus. The posterodorsally oriented base of the ascending ramus of the right maxilla (Figs. 6G, 6H) is preserved, but the anterodorsal margin of the lateral surface is missing. The anteroposterior length of the preserved part of the left maxilla is measuring 28.87 cm, and the right element is 29 cm.

On the lateral surface, the ventral extent of the antorbital fossa is well developed, occupying more than half of the maxillary body ventrally, as in Masiakasaurus (Carrano, Loewen & Sertich, 2011), Marshosaurus (Madsen, 1976b), Allosaurus (Madsen, 1976a), Eocarcharia (Sereno & Brusatte, 2008) and metriacanthosaurids (Currie & Zhao, 1993; Dong, Zhou & Zhang, 1983; Gao, 1999). This is different from the moderate range of the antorbital fossa wall reaching nearly half depth of the maxilla body in Afrovenator (Sereno et al., 1994), Acrocanthosaurus (Eddy & Clarke, 2011), Neovenator (Brusatte, Benson & Hutt, 2008), Concavenator (Cuesta et al., 2018), and Ceratosaurus (Madsen & Welles, 2000). In contrast, the antorbital fossa has very limited exposure on the maxillary body in Torvosaurus (Britt, 1991; Hendrickx & Mateus, 2014), Wiehenvenator (Rauhut, Hübner & Lanser, 2016), Monolophosaurus (Zhao & Currie, 1993; Brusatte et al., 2010a), and some Carcharodontosarurids (Sereno et al., 1996; Coria & Salgado, 1995; Coria & Currie, 2006; Brusatte & Sereno, 2007). Some abelisaurids even totally lack the antorbital fossa on the maxillary body like Majungasaurus (Sampson & Witmer, 2007) and Kryptops (Sereno & Brusatte, 2008). The border of the antorbital fossa is better preserved and well defined by a rim on the left maxilla, while this rim is not evident on the right maxilla due to compression. The antorbital fossa is anteriorly extended, with its anterior-most border reaching the 3rd alveolus, as in Sinraptor (Currie & Zhao, 1993) and Ceratosaurus (Madsen & Welles, 2000), indicating a reduced anterior ramus. Above the 3rd tooth the rim gently curves upward, forming a round anteroventral margin of the antorbital fossa as in Sinraptor (Currie & Zhao, 1993), Yangchuanosaurus (Dong, Zhou & Zhang, 1983), Ceratosaurus (Madsen & Welles, 2000), Marshosaurus (Madsen, 1976b), and Monolophosaurus (Brusatte et al., 2010a). This contrasts with the squared anteroventral border of the antorbital fossa in Eocarcharia (Sereno & Brusatte, 2008), Acrocanthosaurus (Eddy & Clarke, 2011), and Dubreuillosaurus (Allain, 2002). Posteriorly, this rim flattens gradually throughout the posterior ramus.

The preserved ascending ramus of the right maxilla presents the anteroventral margin of the external antorbital fenestra. From the ventral margin of the antorbital fenestra, the preserved ascending ramus is measuring 6.95 cm. The angle between the main axis of the ascending ramus and the jugal ramus of the maxilla is about 60°. The lateral surface of the ascending ramus of the right maxilla is too fragmentary to determine whether it is excavated by pneumatic openings (Figs. 6G, 6H) seen in Sinraptor (Currie & Zhao, 1993; Gao, 1992) and Ceratosaurus (Madsen & Welles, 2000).

Although the ascending rami are largely incomplete on both maxillae, traces of two openings at the base of the ascending ramus are preserved. On the left maxilla, the anterior concavity on the anterodorsal margin of the maxillary body is smooth, and demarcates the ventral rim of an oval natural fenestra, the anterior end of which is adjacent to the anteroventral margin of antorbital fossa. The posterior opening only preserves its rounded ventral half. On the right maxilla, the anterior opening only preserves its posterior rim, while the posterior one is nearly intact. These two openings are respectively interpreted as promaxillary fenestra and maxillary fenestra here based on their relative placement (Witmer, 1997). The preserved portion of the promaxillary fenestra indicates that it is larger than the maxillary fenestra, which resembles the condition in Sinraptor (Witmer, 1997; Hendrickx & Mateus, 2014; Gao, 1992) and Yangchuanosaurus (Dong, Zhou & Zhang, 1983). Relatively large promaxillary fenestra is regarded as a synapomorphy of Metriacanthosauridae (Carrano, Benson & Sampson, 2012). In many other theropods (Neovenator, Dilophosaurus, Dubreuillosaurus: Brusatte, Benson & Hutt, 2008; Marsh & Rowe, 2020; Allain, 2002), the promaxillary fenestra is slit-shaped and covered by a lamina from lateral view. An oval promaxillary fenestra is also presented in Acrocanthosaurus (Eddy & Clarke, 2011), Eocacharia (Sereno & Brusatte, 2008), and some coelurosaurs (Xu et al., 2006; Brusatte et al., 2009). The promaxillary and maxillary fenestrae seem to merge into one opening in Carcharodontosaurinae (Hendrickx & Mateus, 2014; Canale et al., 2014).

The ventral margin of the maxilla is slightly convex, with one row of neurovascular foramina aligning right above and in parallel with it, similar to those presented in Sinraptor (Currie & Zhao, 1993), Allosaurus (Madsen, 1976a), and Monolophosaurus (Brusatte et al., 2010a), in contrast to two rows of neurovascular foramina presented in Marshosaurus (Madsen, 1976b), Shaochilong (Brusatte et al., 2010b), and Eocarcharia (Sereno & Brusatte, 2008). The foramina dorsal to the anterior four alveoli opens anteroventrally, then the orientation of subsequent foramina gradually turns more posteroventrally. Each foramina opens ventrally into a depression but is less extensively than the band-like depression in Ceratosaurus (Madsen & Welles, 2000). Posterior to the 10th alveolus, the foramina merge into a discontinuous groove.

Twelve and 10 functional teeth are preserved on the left and right maxilla, respectively. Based on the vacant space, each maxilla is estimated to bear at least 14 alveoli, similar to the condition in many allosauroids (Currie & Zhao, 1993; Madsen, 1976a; Dong, Zhou & Zhang, 1983). Each tooth is labiolingually compressed and strongly curved backward. Both mesial and distal carinae are serrated, and the well-preserved ninth maxillary tooth of the right maxilla bears 15 and 20 denticles per five mm on the distal and mesial carinae, respectively (Fig. 6D). The distal carina continues to the base of the crown, but the mesial carina reaches less than half the length of the crown from the tip, which is a common condition in theropods. Among these preserved functional teeth, the fourth tooth is the biggest in both left and right maxillae and reaches the axial length of 4.60 cm in left and 3.99 cm in right. The third tooth of the premaxilla, the biggest premaxillary tooth, is similar in size to the first maxillary tooth, which manifests that the size of the teeth is continuous from the premaxilla to the maxilla.

Lacrimal—The right lacrimal is relatively complete (Figs. 7A, 7B), whereas the left one lacks most of its dorsal part (Figs. 4A, 4B). The dorsoventral height of the right lacrimal is 12.08 cm. The ventral ramus of the lacrimal contacts the anterodorsal process of the jugal, and forms most of the anterior rim of the orbit. The ventral process constricts anteroposteriorly at its mid-height, and then expands through the ventral part until it sutures with the jugal. The bone forming the posteroventral corner of antorbital fenestra is broken, so it is not possible to determine whether the lacrimal contacts the maxilla. The angle between the ventral ramus of lacrimal and jugal ramus of the maxilla is 120°, but both of them are disarticulated and displaced from the lacrimal-jugal contact (Figs. 4C, 4D), such a blunt angle might be caused during the preservation. The anterior ramus lacks most of its anterior end, but the posterodorsal margin of antorbital fenestra is preserved. The preserved base of the anterior ramus and ventral ramus meet at an angle slightly more than 90°.

Figure 7 Skull elements of Yuanmouraptorjinshajiangensis gen. et sp. nov. (LFGT-ZLJ0115).

Right lacrimal in (A) lateral view with (B) labeled drawing. Articulated right jugal and quadratojugal in (C) lateral view with (D) labeled drawing. Articulated right jugal and quadratojugal in (E) medial view with (F) labeled drawing. Left jugal in (G) lateral view with (H) labeled drawing. (I) Left quadratojugal and partial quadratojugal ramus of left jugal. Abbreviations: aof, antorbital fossa; de, depression; f, flange; fo, fossa; j, jugal; lc, lacrimal contact; lla, lateral lamina; ltf, lateral temporal fenestra; mla, medial lamina; o, orbit; pn, pneumatic foramen; por, postorbital ramus of jugal; qj, quadratojugal; qjr, quadratojugal ramus of jugal. Striated areas indicate damage. Scale bars represent 50 mm. Photos by Xiao-Chun Wu and Yi Zou.

The ventral ramus is formed by two laminae as in most other tetanurans: a lateral one and a medial one. The lateral lamina protrudes anteriorly into the antorbital fenestra at the 2/3 height of the ventral ramus, and separates the antorbital fossa on the lacrimal into dorsal and ventral part as in Allosaurus (Madsen, 1976a), Monolophosaurus (Brusatte et al., 2010a), and Acrocanthosaurus (Currie & Carpenter, 2000). Although in Torvosaurus (Britt, 1991) the lateral lamina protrudes anteriorly, its anterior-most point does not extend into the antorbital fenestra, resulting the antorbital fossa continuous on anterior and ventral ramus of lacrimal. While in spinosaurids (Charig & Milner, 1997; Schade et al., 2023), the lateral lamina of the lacrimal does not protrude anteriorly.

The posterodorsal part of the lacrimal bears a low, blunt, laterally triangular boss, which is 2.23 cm in length with rugosity distributed on its dorsal and ventral lateral surfaces. This lacrimal boss is proportionally larger than the small boss seen in Torvosaurus (Britt, 1991), but less developed as a horn than in Allosaurus (Madsen, 1976a) and Ceratosaurus (Madsen & Welles, 2000). In Sinraptor (Currie & Zhao, 1993; Gao, 1992; Gao, 1999) and Yangchuanosaurus (Dong et al., 1978; Dong, Zhou & Zhang, 1983), the lacrimal horn is relatively low but bears prominently thicker rugosity than that of Yuanmouraptor. A weak flange is right below the posterodorsal boss of lacrimal, resembling that of Ceratosaurus (Madsen & Welles, 2000) and Sinraptor (Currie & Zhao, 1993). In many carcharodontosaurids (Acrocanthosaurus, Giganotosaurus, Mapusaurus, Meraxes: (Eddy & Clarke, 2011; Coria & Salgado, 1995; Coria & Currie, 2006; Canale et al., 2022), this flange is more pronounced and forms a process, which notably marks the lower limit of the eye socket.

Two pneumatic openings excavate the main posterodorsal body of the lacrimal, located at the posterodorsal rim of antorbital fossa. Besides, a third foramen is about 0.8 cm below the larger posterior opening, and falls within the region of antorbital fossa, similar to Allosaurus (Madsen, 1976a). But this foramen differs from the opening of nasolacrimal conduct presented in Abelisaurids (Sampson & Witmer, 2007; Cerroni, Canale & Novas, 2020), in which the opening is unobservable in lateral view.

Jugal—The anterodorsal border of the left and right jugals are broken, so it is unclear whether the jugal separates the maxilla and lacrimal and slightly contributes to the antorbital fenestra. The preserved left jugal is 17.42 cm long (Figs. 7G, 7H) and the right one is 16.40 cm (Figs. 7C–7F ). The anterior ramus of the left jugal rises dorsally into the lacrimal ramus to contact the lacrimal, and contributes to the anteroventral rim of orbit. The postorbital ramus of the jugal contributes to the posteroventral margin of the orbit, and at its dorsal tip the jugal reaches the dorsoventral height of 7.02 cm. The postorbital ramus is vertically oriented and forms a steep angle with the anterior ramus. These two rami result in an acute ventral margin of the orbit, similar to the condition in Sinraptor (Currie & Zhao, 1993), Yangchuanosaurus (Dong, Zhou & Zhang, 1983), and Allosaurus (Madsen, 1976a). Beneath the ventral rim of the orbit, the dorsoventral depth of jugal is 3.3 cm on the left and 2.91 cm on the right. The ventral margin of both right and left jugals is relatively smooth, differing from the abruptly changed orientation occurs in Allosaurus fragilis (Madsen, 1976a).

Posteriorly, the quadratojugal ramus of the jugal bifurcates into an upper branch overlapping the anterior ramus of the quadratojugal and a lower branch lying below the quadratojugal as in most theropods, but differs from the triradiate posterior ramus of Sinraptor (Currie & Zhao, 1993). In the better-preserved right jugal (Fig. 7D), the upper branch is slightly shorter than the lower branch, which differs from the much-shortened upper branch seen in Allosaurus (Madsen, 1976a) and Monolophosaurus (Brusatte et al., 2010a). The quadratojugal ramus strongly turns upwards on the right jugal, and results in the convex ventral rim of lateral temporal fenestra. This exaggerating curvature is more likely the distortion caused by compression. In contrast, on the left jugal, the quadratojugal ramus curves slightly downwards near the tip of the upper and lower branches (Figs. 7G, 7H), which also might be the consequence of distortion.

The posteroventral rim of the antorbital fossa is well developed on the jugal, and the rim is demarcated by a ridge which continues onto the ventral ramus of the lacrimal. At the base of this ridge, the lateral surface of the jugal is smooth, and differs from the pneumatic openings seen in more derived metriacanthosaurids such as Sinraptor (Currie & Zhao, 1993; Gao, 1992; Gao, 1999) and Yangchuanosaurus (Dong et al., 1978; Dong, Zhou & Zhang, 1983). Beneath the postorbital ramus, near the bottom of the left jugal, the lateral surface is penetrated by a small and flat foramen (Fig. 7H), but this foramen is absent on the right jugal. This might be considered as break, but a similar foramen is also presented in Allosaurus (Madsen, 1976a) and Sinraptor (Currie & Zhao, 1993).

Quadratojugal—Both the left and right quadratojugals are preserved, the left one (Figs. 7C–7F) is largely incomplete on its lateral surface, while the right one (Fig. 7I) lacks most of its dorsal ramus. The quadratojugal is L-shaped in lateral view, and compressed mediolaterally as in most theropods. The left quadratojugal is 11.88 cm long and 6.48 cm high, while the right one is 12.96 cm long and 0.73 cm thick.

In lateral view, the ventral margin of the quadratojugal is convex, similar to the condition in Ceratosaurus (Madsen & Welles, 2000) and Sinraptor hepingensis (Gao, 1992; Gao, 1999). The posterior end of the bone forms a triangular process oriented posteriorly. The lateral surface of the quadratojugal is smooth, with a slight depression (Figs. 7C, 7D) extending throughout the base of the dorsal ramus and occupy roughly 2/3 ventral depth of the main body, similar to the rounded fossa in Meraxes (Canale et al., 2022). The anterior process tapers anteriorly and is wedged into the upper and lower branches of the quadratojugal ramus of the jugal. The anterior process extends to be level with the anterior border of the lateral temporal fenestra, more anteriorly than those of Allosaurus (Madsen, 1976a) and Sinraptor dongi (Currie & Zhao, 1993), but falls shorter than the condition in Monolophosaurus (Brusatte et al., 2010a). The dorsal process is preserved on the left quadratojugal, but most of its external surface is broken. The preserved dorsal process takes the form of triangle, and tapers dorsally, similar to that of Alpkarakush (Rauhut et al., 2024), but in the latter the posterior part of the quadratojugal protrudes less posteriorly than in Yuanmouraptor. The articulation with the squamosal is not definitive due to the missing of the dorsal tip.

In the medial view, the posterior end of the quadratojugal bears slight rugosity, which is the contact with the lateral condyle of the quadrate as in Allosaurus (Madsen, 1976a) and Alpkarakush (Rauhut et al., 2024). Anterodorsal to this rugosity, a deep fossa which excavate the medial surface and is bordered posteriorly by a rounded rim, together with the aforementioned concave lateral surface result in a thin lamina in this region (Figs. 7E, 7F).

Postorbital—Only the right postorbital has been preserved, and lacks most distal part of its posterior ramus (Figs. 8A, 8B). In lateral view, the postorbital is T-shaped in outline consisting of the orbital, posterior and ventral rami. The postorbital measures 9.83 cm in dorsoventral height and 6.17 cm in anteroposterior length from the orbital ramus to the broken base of the posterior ramus.

Figure 8 Postorbital and skull roof of Yuanmouraptorjinshajiangensis gen. et sp. nov. (LFGT-ZLJ0115).

Right postorbital in (A) lateral view with (B) labeled drawing, and in (C) posterior view with (D) labeled drawing. Skull roof in (E) dorsal view with (F) labeled drawing. Abbreviations: de, depression; f, frontal; l, lacrimal; la, lamina; lsp, laterosphenoid; n, nasal; or, orbital ramus; ot, otoccipital; pa, parietal; po, postorbital; pop, paroccipital process; por, posterior ramus; pr, prootic; prf, prefrontal; r, ridge; so, supraoccipital; stf, supratemporal fenestra; stfo, supratemporal fossa; t, trough; vr, ventral ramus of postorbital. Striated area indicates damage and grey area indicates matrix. Scale bars represent 50 mm. Photos by Xiao-Chun Wu.

The postorbital projects anteriorly to form a sheet-shaped process (Figs. 8A, 8B), differing from the prominent orbital boss seen in other derived metriacanthosaurids (Dong et al., 1978; Dong, Zhou & Zhang, 1983; Gao, 1992; Currie & Zhao, 1993; Rauhut et al., 2024). From the juncture of the orbital ramus and ventral ramus of the postorbital, the orbital ramus is 2.69 cm long. Through this planar orbital ramus, the postorbital contacts the frontal medially, and forms the orbital roof along with the prefrontal, lacrimal and a slight part of the frontal as in Sinraptor (Currie & Zhao, 1993) and Allosaurus (Madsen, 1976a). In contrast, the frontal or prefrontal is excluded from the orbital rim due to the postorbital-lacrimal articulation in carcharodontosaurids (Sereno et al., 1996; Sereno & Brusatte, 2008; Cuesta et al., 2018; Canale et al., 2022). The orbital ramus maintains a relatively constant thickness with the deepest portion measured 0.99 cm. This constant thickness of the orbital process of postorbital in Yuanmouraptor also differs from conditions in Torvosaurus (Britt, 1991) and Eustreptospondylus (Sadleir, Barrett & Powell, 2008), in which the orbital processes increase the dorsoventral depth gradually backwards.

The ventral ramus tapers downward in lateral view. The posterior part of the ventral ramus constricts in transverse width to form a prominent lamina (Figs. 8A–8D). This lamina runs along the posterior rim of the ventral ramus, and is positioned near its midline, while in Sinraptor (Currie & Zhao, 1993) and Alpkarakush (Rauhut et al., 2024) this lamina is placed more laterally. In contrast, such lamina is not well developed in many megalosauroids (Torvosaurus, Eustreptospondylus, Wiehenvenator: Britt, 1991; Sadleir, Barrett & Powell, 2008; Rauhut, Hübner & Lanser, 2016), and a prominent U-shaped groove runs along nearly half the ventral part of the ventral ramus in posterior view. Through the ventral part of this lamina the postorbital contacts the postorbital ramus of the jugal. The upper half of the ventral ramus extends posteroventrally, then the lower half turns downwards with its tip curves backwards, resulting in a gently sigmoidal profile in lateral view. The anterior rim of the ventral ramus is smooth and concave, and there is no evidence of any anteriorly projecting intraorbital process which defines the ventral border of eyeball seen in Carcharodontosaurus (Sereno et al., 1996) and abelisaurids like Majungosaurus (Sampson & Witmer, 2007) and Carnotaurus (Bonaparte, Novas & Coria, 1990). A shallow trough begins at the anterodorsal rim of the orbit ramus, then twists to face laterally on the ventral ramus and shallows ventrally, which is considered as an autapomorphy of Yuanmouraptor. The bone surface in the region of this trough bears slight rugosity.

The roof of the postorbital body expands laterally into a longitudinal ridge. This longitudinal ridge continues onto the posterior ramus of postorbital, and marks the lateral rim of the supratemporal fossa. In the lateral view, ventral to the ridge the lateral surface of main body forms a shallow depression (Fig. 8B), which is similar to the condition in Sinraptor (Currie & Zhao, 1993), Allosaurus (Madsen, 1976a), Alpkarakush (Rauhut et al., 2024), Eocarcharia (Sereno & Brusatte, 2008), and Wiehenvenator (Rauhut et al., 2024). Whereas in derived carcharodontosaurids (Brusatte & Sereno, 2007; Sereno & Brusatte, 2008), this depression is absent. The preserved posterior ramus is mediolaterally thin, and its cross section tapers dorsolaterally to form the ridge (Figs. 8C, 8D). This differs from Sinraptor (Currie & Zhao, 1993) and Alpkarakush (Rauhut et al., 2024), in which the posterior ramus of the postorbital is dorsoventrally compressed, resulting a plate instead of a ridge in dorsal view. The posterior ramus is deflected at an angle of nearly 70° from the posteroventrally pointed ventral ramus.

In dorsal view (Figs. 8E, 8F), a medial process contacts frontal anteriorly and laterosphenoid posteromeidally, forms the anterolateral border of the supratemporal fenestra. Between the frontal and laterosphenoid, this process also has a very limited contact with the lateral projection of the parietal similar to the condition of Sinraptor (Currie & Zhao, 1993). The supratemporal fossa is shown as a shallow but well-defined depression, and its anterior rim is formed by the postorbital together with the frontal, parietal and laterosphenoid.

Prefrontal— The prefrontal is a small element located between the lacrimal and frontal, partly lacks its anterior end (Figs. 8E, 8F). The preserved prefrontal measures 5.15 cm in lenghth, 2.70 cm in width and 1.88 cm in depth. Due to the damage, the prefrontal is displaced posteriorly relative to the frontal, and overlaps the mediodorsal surface of the lacrimal. The nasal is poorly preserved, thus the articulation between the prefrontal and nasal is not definitive.

In dorsal view, the prefrontal is sub-rhomboid in outline, and contacts the lacrimal laterally and the frontal medially, but sutures of these articulations are all broken. The dorsal surface of prefrontal is planar and smooth. The posterolateral rim of the prefrontal contributes to the orbital roof and is slightly rugose as in Sinraptor (Currie & Zhao, 1993), this rugosity might continue onto the lacrimal boss. Unlike the fusion with other bones or lateral covering of the lacrimal in carcharodontosaurids (Sereno et al., 1996; Sereno & Brusatte, 2008) and Majungasaurus (Sampson & Witmer, 2007), the prefrontal of Yuanmouraptor is exposed laterally on the dorsal rim of the orbit and forms a supraorbital notch together with the frontal, postorbital and lacrimal, a condition close to that in Allosaurus (Madsen, 1976a), Sinraptor (Currie & Zhao, 1993), and Monolophosaurus (Brusatte et al., 2010a).

Frontal—The paired frontals (Figs. 8E, 8F) are wedge-shaped in dorsal view, and articulate each other through the suture on the midline, though the structure of this suture is deformed due to the compressional distortion. Both left and right frontals are preserved, but lack their anterior end, thus the articular surface with nasal is not definitive. The right frontal is relatively complete, about twice as long as it is wide, and measures 10.82 cm in length and 5.77 cm in width. Whereas the left frontal lacks its posterolateral part.

In dorsal view, the frontal contacts the prefrontal anterolaterally the postorbital posterolaterally, and the parietal posteromedially. The frontal reaches its greatest mediolateral width at a point level with its contact with the postorbital, resembling that in Sinraptor (Currie & Zhao, 1993), Allosaurus (Madsen, 1976a), Acrocanthosaurus (Currie & Carpenter, 2000), and Carcharodontosaurus (Sereno et al., 1996). In contrast, the frontal of Eustreptospondylus (Sadleir, Barrett & Powell, 2008) is widest at the supraorbital notch. Prior to the contact with the postorbital, the transverse width of the frontal shrinks abruptly to contribute to the supraorbital notch, this also occurs in Sinraptor (Currie & Zhao, 1993) and Allosaurus (Madsen, 1976a). At the anterior rim of the supraorbital notch, the frontal expands laterally to form the contact with the prefrontal, then the frontal tapers anteriorly. The dorsal surface of the frontal is smooth, and its posterolateral part was occupied by a well-defined shallow recess, which is continuous with the recess on the postorbital and contributes to the anterior rim of supratemporal fossa as in Sinraptor (Currie & Zhao, 1993), Allosaurus (Madsen, 1976a), and Eocarcharia (Sereno & Brusatte, 2008). Whereas in derived carcharodontosaurids (Sereno et al., 1996; Coria & Currie, 2006) the occupation of the supratemporal fossa is strongly reduced. Although the contact between the paired frontals is broken, there is no sign of a midline ridge displayed in Shaochilong (Brusatte et al., 2010b). Posteriorly, the suture with the parietal is interdigitating medially and roughly straight laterally, resembles that of Sinraptor (Currie & Zhao, 1993), Allosaurus (Madsen, 1976a) and Shidaisaurus (Wu et al., 2009).

Parietal—The paired parietals are fused, and lack most of their posterodorsal part (Figs. 8E, 8F). The preserved parietal is similar to that of Sinraptor (Currie & Zhao, 1993) and Shidaisaurus (Wu et al., 2009) in outline, and measures 6.69 cm in length and 8.24 cm in width. In dorsal view, the parietal contacts the frontal through an interdigitating suture anteriorly. In contrast, this suture forms a ridge in tyrannosaurids (Currie, 2003, Brusatte et al., 2009). Posterior to this suture with the frontal, the parietal expands laterally to form two slender projections, contacting the frontal anteriorly through a pair of transversely extended sutures. The tips of these projections reach the postorbital and overlap the laterosphenoid. The dorsal surface between the projections is dorsally convex, and differs from the additional bone deposition which protrudes laterally into the supratemporal fossa seen in Sinraptor (Currie & Zhao, 1993). Posterior to these projections, the parietal constricts transversally and measures as 2.84 cm in width. Due to the damage presented on the posterodorsal part of the parietal, the existence of the nuchal crest and the border between the parietal and supraoccipital is not definitive, as well as the border of the supratemporal fossa on the dorsal surface of the parietal.

Squamosal—Only the right squamosal has been preserved (Figs. 9A–9E), but deviates strongly from the original position, with its anterior and ventral end obscured by sediment. As in many theropods, the squamosal of Yuanmouraptor comprises four processes: the anterior process contacting the posterior ramus of the postorbital; the ventral process that extends ventrally to cover the quadrate laterally and contact the ascending process of the quadratojugal; the medial process underlapping the parietal; and a posterior process that envelopes the paroccipital process anterolaterally.

Figure 9 Skull elements of Yuanmouraptorjinshajiangensis gen. et sp. nov. (LFGT-ZLJ0115).

Right squamosal in (A) dorsal view with (B) labeled drawing, in (C) posterodorsal view with (D) labeled drawing, and in (E) lateral view. Left quadrate in (F) anterolateral, (G) posterolateral, and (H) ventral views. Left palatine in (I) lateral view with (J) labeled drawing. Abbreviations: ap, anterior process; ecc, ectocondyle; enc, entocondyle; ics, intercondylar sulcus; in, internal naris; ltf, lateral temporal fenestra; mp, medial process; map, maxillary process; pac, parietal contact; po, postorbital; popc, paroccipital process contact; pn, pneumatic foramen; pp, posterior process; qc, contact for quadrate; qs, quadrate shaft; r, ridge; vp, ventral process; vptp, vomeropterygoid process. Striated area indicates damage and grey area indicates matrix. Scale bar represents 50 mm. Photos by Xiao-Chun Wu and Yi Zou.

In dorsal view (Figs. 9A, 9B), the preserved part of the squamosal measures 6.24 cm in maximum length and 6.62 cm in width. The dorsal surface of the squamosal is smooth and slightly concave, and subtriangular in outline. The obscuration of the postorbital and sediment precludes the observation of the posterolateral rim of the supratemporal fenestra, which is formed by the medial rim of the squamosal in dorsal view. The lateral rim of the dorsal surface is demarcated by a ridge (Fig. 9B) which extends posteriorly from the anterior process then recurves medially at the base of the posterior process. The articular surfaces for the parietal and the paroccipital process together form the posterior border of the squamosal (Fig. 9D). The medial process protrudes anteromedially and is slightly convex posteromedially. The coarse surface of the medial process, which is not continuous with the dorsal surface of the bone, indicates its contact with the parietal.

In lateral view (Fig. 9E), the preserved anterior process measures 3.36 cm in length, and bears evident dorsal and ventral rim, and the former contributes to the border of the concavity on the dorsal surface. The preserved ventral process measures 3.25 cm in depth and inclines anteroventrally. Together the anterior and ventral processes of the squamosal form the posterodorsal rim of the lateral temporal fenestra, these two processes are at an angle of broadly 70°. In contrast, this angle is blunter in basal-branching tetanurans such as Sinraptor (Currie & Zhao, 1993), Allosaurus (Madsen, 1976a), Afrovenator (Sereno et al., 1994), and Eustreptospondylus (Sadleir, Barrett & Powell, 2008). The tip of the posterior process projects 1.89 cm posteriorly from the main body of the squamosal with posteriorly tapering outline, which differs from the posteroventrally oriented and prominently expanded posterior process in Acrocanthosaurus (Eddy & Clarke, 2011), Allosaurus (Madsen, 1976a) and Monolophosaurus (Brusatte et al., 2010a). The lateral surface of the posterior process of the squamosal is rugose and pitted, might be for the attachment of ligament. In basal tetanuran such as Monolophosaurus (Brusatte et al., 2010a), derived allosauroids such as Acrocanthosaurus (Eddy & Clarke, 2011) and Allosaurus (Madsen, 1976a), and basal coelurosuar Zuolong (Choiniere et al., 2010), there is a ventrally faced concavity located between the posterior process and the ventral process, allowing lateral exposure of the quadrate head. Whereas in Yuanmouraptor and other metriacanthosaurids (Dong, Zhou & Zhang, 1983; Currie & Zhao, 1993; Gao, 1992; Gao, 1993), the quadrate head is obscured laterally by the posterior process of the squamosal, which is considered as a synapomorphy of Metriacanthosauridae (Carrano, Benson & Sampson, 2012).

In posterior view (Figs. 9C, 9D), the squamosal is dorsoventrally deep and posteriorly concave along the contact with the paroccipital process, and reaches its maximum depth near the posterior process. Then the posterior surface of the squamosal becomes dorsoventrally shallower and flat towards its medial process. The partly exposed ventral surface of the posterior process is concave ventrally for housing the quadrate head.

Quadrate—Both the left and right quadrates are poorly preserved. The left quadrate (Figs. 9F–9H) lacks most of its dorsal part and the right one (Fig. 4D) only preserves partial pterygoid flange. The ventral part of the left quadrate is well preserved, with measuring 4.69 cm in mediolateral width, and is separated by an anteromedially oriented intercondylar sulcus into the entocondyle and the ectocondyle (Fig. 9H). The ectocondyle is larger, which is similar to the condition in Allosaurus (Madsen, 1976a), but contrasts with that of Eustreptospondylus (Sadleir, Barrett & Powell, 2008) and Ceratosaurus (Madsen & Welles, 2000), in which the entocondyle is rather larger. The ectocondyle is 2.79 cm wide mediolaterally while the entocondyle is 2.5 cm wide. In ventral view, the long axes of the ento- and ectocondyles are anteromedially oblique in a broadly same direction, but slightly more medially directed than the degree of the intercondylar sulcus (Fig. 9H). The anterior end of each condyle is at approximately the same level, which contrasts with the strongly anterior protruding entocondyle in Torvosaurus (Britt, 1991) and Eustreptospondylus (Allain, 2002). In lateral view, both mandibular condyles extend anteriorly to form a concavity dorsal to them, while the posterior rim of the preserved quadrate is straight along the shaft.

Palatine—The incomplete left palatine is preserved (Figs. 9I, 9J) and lacks most of its posterior processes, with its medial surface being obscured by matrix. Whereas the right palatine is presented by a bar-like bone, and its structure is not possible to identify. The palatine takes the form of a saddle, with the central part of the bone being lowest in dorsoventral depth. The preserved palatine is 11.23 cm long anteroposteriorly, 4.5 cm deep dorsoventrally at the vomeropterygoid process, and 2.64 cm deep at the waisted region of the main body.

The anterior vomeropterygoid and maxillary processes are preserved, together define the posterior limit of the internal naris choana. The vomeropterygoid process lacks its distal end, and inclines mediodorsally at its base, then becomes more anteriorly oriented, thus the medial rim of the coana is concave laterally. The maxillary process is more robust than the vomeropterygoid process, and extends anteriorly with a laterally convex surface. The articular surface with the maxilla of the palatine is not definitive due to the damage. In lateral view, a shallow fossa is presented at the base of the maxillary process, immediately posterior to the internal naris choana. This fossa might mark the pneumatic recess of palatine, but does not penetrate the surface of bone and lead to the inner space presented in Sinraptor (Currie & Zhao, 1993) and Acrocanthosaurus (Eddy & Clarke, 2011). A similar fossa is presented in Neovenator (Brusatte, Benson & Hutt, 2008), but it is located more posteriorly at the juncture of the jugal and medial processes.

Supraoccipital—The supraoccipital is poorly preserved, with most of its dorsal part missing. The preserved part of the supraoccipital (Figs. 10A, 10B) is 2.73 cm deep dorsoventrally and 5.19 cm wide transversely at its ventral part. Despite the incompleteness, the preserved ventral part of the supraoccipital indicates a prominent ridge running along the midline, and flanked by a pair of sub-vertical grooves. Lateral to the paired grooves, the bone extends posterolaterally and might continue onto the probable nuchal crest of parietal. The ventral part of the middle ridge is triangular in shape, tapering dorsally, and expands transversely based on the dorsal fracture. The ventral rim of the supraoccipital makes a slight contribution to the dorsal border of the foramen magnum as in Sinraptor (Currie & Zhao, 1993), Acrocathosaurus (Eddy & Clarke, 2011), Monolophosaurus (Zhao & Currie, 1993), and Piatnitzkysaurus (Rauhut, 2004), and this contribution measures 0.83 cm in breadth. However, the supraoccipital is excluded from the foramen magnum by medially contact of the exoccipitals in Shidaisaurus (Wu et al., 2009). On the left side of the ventral part of the bone, there are two foramina that penetrate the external surface, but on the opposite side the surface is smooth. The symmetrical paired foramina positioned lateral to the midline near the ventral part of the supraoccipital are generally referred as exits for external occipital vein (Currie & Zhao, 1993), vena capitis dorsalis (Coria & Currie, 2002; Rauhut, 2004; Brusatte & Sereno, 2007; Eddy & Clarke, 2011) or vena cerebralis media (Sampson & Witmer, 2007). The asymmetrical foramina presented in the supraoccipital of Yuanmouraptor might be caused by pathology or taphonomic process. The supraoccipital expands laterally at its ventral margin to contact the otoccipitals.

Figure 10 Occiput of Yuanmouraptor jinshajiangensis gen. et sp. nov. (LFGT-ZLJ0115).

Supraoccipital and adjacent bones in (A) posterodorsal view with (B) labeled drawing. (C) Depression housing cranial nerves. Braincase in (D) posterior view with (D) labeled drawing. Abbreviations: bo, basioccipital; bs, basisphenoid; bt, basal tuber; eo, exoccipital; fm, foramen magnum; oc, occipital condyle; op, opisthotic; pop, paroccipital process; r, ridge; so, supraoccipital; vcd, foramen vena capitis dorsalis; X, XI, XII, foramina for cranial nerve X, XI, XII; ?, unknown bone. Striated area indicates damage. Scale bars represent 50 mm. Photos by Xiao-Chun Wu.

Otoccipital (Exoccipital-Opisthotic)—The main body of the exoccipital-opisthotic complex is well preserved, but the tips of the left and right paroccipital processes are missing (Figs. 10D, 10E). A possible suture between the exoccipital and opisthotic is presented. Given that in all preserved vertebrae, the neural arch is attached to the centrum and the neurocentral suture is absent in most of them, indicating that this individual is nearly mature or subadult. Thus this suture-like boundary on the exoccipital-opisthotic complex is more likely caused by damage.

In posterior view (Figs. 10D, 10E), the paired exoccipitals are separated by the supraoccipital above the foramen magnum. Then the exoccipitals form the lateral and ventral margin of the foramen magnum, and meet each other at a midline suture throughout the dorsal surface of the occpital condyle. The foramen magnum is 2.47 cm transversely wide and 1.32 cm dorsoventrally high, proportionally broader than that of Sinrapor (Currie & Zhao, 1993) and Allosaurus (Madsen, 1976a). The paroccipital process is posterolaterally directed, and slightly turns downwards, which contrasts with more sharply downturned condition in Sinrapor (Currie & Zhao, 1993) and Allosaurus (Madsen, 1976a). The ventral limit of the base of the paroccipital process levels with the bottom of the occipital condyle, as in Allosaurus, Sinraptor, and Piatnitzkysaurus (Rauhut, 2004), but contrasts with more dorsally placed ventral base of the paroccipital process in Dubreuillosaurus (Allain, 2002), Eustreptospondylus (Sadleir, Barrett & Powell, 2008), and Leshansaurus (Li et al., 2009). A depressed area (Fig. 10C) lies between the paroccipital process and the base of the occipital condyle, and houses three foramina for cranial nerves. Among these foramina, the dorsal one is for the vagus (X) and accessory (XI) cranial nerves, the medioventral one and the lateroventral one is for the two branches of the hypoglossal nerve (XII) (Fig. 10C). The ventral part of the exoccipital-opisthotic tapers ventrally, and overlaps the basioccipital laterally at the boundary between the basioccipital and basisphenoid. The suture with the basioccipital extends from the base of the occipital condyle to the basal tubera.

In lateral view (Figs. 11A, 11B), the anterodorsal corner of the exoccipital-opisthotic is overlapped by the prootic, and the exoccipital-opisthotic forms the posterior boundary of the fenestra ovalis approximately ventral to the crista prootica. The posteroventral rim of the paroccipital process, formed by the metotic strut, is strongly concave, and separates the lateral and posterior surfaces of the braincase. The suture with the basisphenoid is posteroventrally inclined and slightly posteriorly concave.

Figure 11 Braincase of Yuanmouraptor jinshajiangensis gen. et sp. nov. (LFGT-ZLJ0115).

Braincase in (A) lateroposterior view with (B) labeled drawing and in (C) laterodorsal view with (D) labeled drawing. Abbreviations: bo, basioccipital; bs, basisphenoid; bt, basal tuber; cfp, cultriform process; cp, crista prootica; fo, fenestra ovalis; lsp, laterosphenonid; oc, occipital condyle; op, opisthotic; pa, parietal; po, postorbital; pop, paroccipital process; pp, prootic pendant; pr, prootic; vcd, foramen vena capitis dorsalis; VII, cranial nerve VII (facial nerve); so, supraoccipital. Straited area indicates damage and grey area indicates matrix. Scale bars represent 50 mm. Photos by Xiao-Chun Wu.

Prootic—The prootic is mainly exposed on the lateral surface (Fig. 11), and the right prootic is better preserved than the left one, with a complete prootic pendant. The prootic is situated posterior to the laterosphenoid. The lateral surface of the prootic is shallowly recessed. Ventral to the suture between the prootic and laterosphenoid, a longitudinal groove runs through the ventral part of the prootic. Approximately posterior to the groove, a single foramen penetrating the bone houses the cranial nerve VII, and differs from the condition of two openings for the cranial nerve VII in Eustreptospondylus (Sadleir, Barrett & Powell, 2008). The opening for the trigeminal (V) nerve originates in the anterior-most part of the prootic and is even bounded by the laterosphenoid anteriorly in many tetanurans such as Sinraptor (Currie & Zhao, 1993), Eustreptospondylus (Sadleir, Barrett & Powell, 2008), Dubreuillosaurus (Allain, 2002), Monolophosaurus (Brusatte et al., 2010a), and Piatnitzkysaurus (Rauhut, 2004). but in Yuanmouraptor, this part is obscured by the sediment, preventing further observation.

The prootic contacts the exoccipital-opisthotic posteroventrally with a jagged suture. A prominent crista prootica (Fig. 11B) marks the posteroventral margin of the prootic. Dorsal to the crista prootica, the prootic is contiguous with the base of paroccipital process and overlaps the exoccipital-opisthotic laterally. The prootic forms the anterior boundary of the fenestra ovalis with the exoccipital-opisthotic forming its posteroventral boundary. And the fenestra ovalis is positioned approximately ventral to the crista prootica. Anteroventral to the fenestra ovalis, a slight process sits at the boundary between the exoccipital-opisthotic and basisphenoid, and protrudes posteroventrally, similar to that in Sinraptor (Currie & Zhao, 1993).

The ventral part of the prootic mainly overlaps the basisphenoid laterally through the prootic pendant, and the space between the pendant and exoccipital-opisthotic is filled sediment.

Basioccipital—The basioccipital is mainly exposed in posterior view (Figs. 10D, 10E), and its central ventral part is incomplete. The basioccipital occupies more than 60 percent of the occipital condyle. The occipital condyle is evidently wider (3.28 cm) than tall (2.3 cm), with rounded and smooth articular surface. Unlike in Sinraptor (Currie & Zhao, 1993), Allosaurus (Madsen, 1976a), and Eustreptospondylus (Sadleir, Barrett & Powell, 2008), the basioccipital of Yuanmouraptor does not contribute to the foramen magnum, and the paired exoccipitals meet each other in the midline. Ventral to the neck of the occipital condyle, a shallow fossa is presented, and the surface ventral to this fossa is posteriorly concave and smooth, differing from the well-defined groove in Piatnitzkysaurus (Rauhut, 2004). The ventral rim of the basioccipital on the right side probably represents the basal tuber, though it is not very apparent due to the damage. The preserved basal tuber is level with the ventral limit of the exoccipital-opisthotic, in contrast to the unusual condition of Sinraptor (Currie & Zhao, 1993), in which the exoccipital-opisthotic extends significantly more ventrally than the basal tubera. Differing from the relatively narrow width between the basal tubera in Sinraptor (Currie & Zhao, 1993), Allosaurus (Madsen, 1976a), and Monolophosaurus (Zhao & Currie, 1993), the transverse width across the basal tubera is broader than the transverse diameter of the occipital condyle in Yuanmouraptor. The suture with the basisphenoid is visible near the basal tubera and could only be observed in right lateral view.

Basisphenoid—The basisphenoid could be mainly observed on the right side (Fig. 11). Its ventral structures such as the basisphenoid recess and the basipterygoid process are obscured by other bones and sediment. In lateral view, the basisphenoid contacts the exoccipital-opisthotic through a posteriorly curved suture. The suture with the basioccipital is visible on the tip of the basal tubera. The dorsal part of the basisphenoid is overlapped by the prootic. Only the base of the cultriform process is exposed, and the anterior part of it is obscured by matrix.

Laterosphenoid—Both the left and right laterosphenoids are preserved, but most of their ventral parts are either damaged or obscured by matrix. The laterosphenoid forms the anterior wall of the braincase, and is surrounded dorsally by the parietal, posteriorly by the prootic and laterally by the postorbital. The contact with the frontal is not definitive due to the bloking of the surrounding articulated bones. The laterosphenoid is visible in dorsal and laterodorsal views (Figs. 8 and 11), and forms the anteromedial rim of the supratemporal fenestra.

Dentary—Both the left and right dentaries are preserved, but their posterior boundaries are broken. The relatively complete right dentary (Figs. 4C, 4D) is 34.1 cm long and reaches minimum depth (4.17 cm) at the level of the fourth alveolus. In lateral view, the main part of the upper margin is straight, but a step appears at nearly the fourth dentary teeth leading to a slight dorsoventral expansion as in Eustreptospondylus (Sadleir, Barrett & Powell, 2008). In contrast to the square anteroventral rim of dentary in Giganatosaurus (Coria & Salgado, 1995), the tip of the ventral rim of dentary in Yuanmouraptor is rounded. The lower margin of the dentary is concave and inclines more ventrally at the 11th alveolus, posterior to which the dentary body expands dorsoventrally throughout its posterior half. An array of slightly undulate neurovascular foramina (Figs. 12A, 12B) excavates the external surface of the bone below anterior seven teeth. Posterior to and level with these foramina, a longitudinal groove extends from the 11th alveolus along the rest dentary and runs upward gradually. Several smaller foramina are scattered over the anteroventral margin of the lateral surface.

Figure 12 Mandibular elements of Yuanmouraptorjinshajiangensis gen. et sp. nov. (LFGT-ZLJ0115).

Left dentary in (A) lateral view with (B) labeled drawing. Left posterior part of mandible in (C) lateral view with (D) labeled drawing. Right posterior part of mandible in (E) medial view with (F) labeled drawing. Abbreviations: af, adductor fossa; an, angular; ap, angular process; ar, articular; ct, foramen of chorda tympani; d4-14, dentary teeth 4-14; emf, external mandibular fenestra; g, groove; hp, hook like process of surangular; imf, internal mandibular fenestra; lg, lateral glenoid; mame, attachment of M. adductor mandibulae externus; nf, neurovascular foramina; par, prearticular; psf, posterior surangular foramen; retp, retroarticular process; sa, surangular; sp, splenial; sr, surangular ridge. Striated area indicates damage. Scale bar represents 50 mm. Photos by Xiao-Chun Wu and Yi Zou.

Most part of the left and the right dentaries adheres to the matrix or other bones, only a bit of medial surface of the left dentary is observable but poorly preserved (Fig. 4D). Through this limit exposure of medial surface, two unfused interdental plates are preserved on the exposed medial surface, and takes the form of sub-triangle, as in Sinraptor (Currie & Zhao, 1993), Dubreuillosaurus (Allain, 2002), Marshosaurus (Madsen, 1976b), and Eustreptospondylus (Sadleir, Barrett & Powell, 2008). Ventral to the interdental plates, a trough represents the paradental groove, which demarcates the ventral border of the interdental plates. A replacement tooth occurs between these two interdental plates, and an unerupted tooth is exposed on the broken surface near the damaged interdentary symphysis, exhibiting serrated distal carina. The Meckelian groove appears as a narrow trough on the preserved dentary, and anterior to which the similar foramina seen in Sinraptor (Currie & Zhao, 1993) and Allosaurus (Madsen, 1976a) might be damaged.

The relatively well-preserved left dentary (Figs. 12A, 12B) bears nine functional teeth, from gaps among which, at least 14 alveoli are estimated. A total of 14–17 teeth are presented in other derived Late Jurassic allosauroids such as Sinraptor (Currie & Zhao, 1993), Yangchuanosaurus (Dong, Zhou & Zhang, 1983), and Allosaurus (Madsen, 1976a). Similar to the condition in maxillary teeth, the distal carina of dentary teeth continues form the base to the tip while the mesial carina develops along less than half of the teeth from the apex. The dentary teeth are generally smaller in size than maxillary tooth with the largest dentary teeth measures 2.73 cm high. Dentary teeth also have greater curvature than those of maxillary and premaxillary teeth.

Splenial—Only the right splenial is observable, and is poorly preserved, with most of its anterior part obscured by other elements and the posterior boundary being damaged (Figs. 12E, 12F). Caused by the compression during burial, the bone covers the anterior end of the prearticular medially. The bone forms a posteroventrally tapering process and its anteroventral rim is slightly concave and smooth.

Surangular—The left surangular (Figs. 12C, 12D) is better preserved than the right one, which is heavily compressed dorsoventrally, and the anterior end of each one is obscured or damaged (Fig. 4). The right surangular measures 27.26 cm anteroposteriorly, and its suture with the dentary is not clear due to the compression.

Anterolateral to the contact with the articular, a longitudinal surangular ridge (Fig. 12D) is presented approximately below the dorsal margin of the surangular. The ridge ends posteriorly with a dorsal concavity, which laterally demarcates the lateral glenoid contacting the ectocondyle of the quadrate. Posterior to the lateral glenoid, the surangular forms a U-shaped notch dorsoventrally deeper than the lateral glenoid. This notch contributes to the retroarticular process together with the articular, and is similar to but is dorsoventrally excavated more deeply than that of Sinraptor (Currie & Zhao, 1993) and Acrocanthosaurus (Eddy & Clarke, 2011). The posterolaterally opened posterior surangular foramen is exhibited posteroventrally to the lateral surangular ridge and ventrolaterally to the mandibular joint. This contrasts with the condition in Monolophosaurus (Brusatte et al., 2010a), in which the lateral surangular ridge is absent and the posterior surangular foramen is ventral to an unexpanded and smooth surface. In addition, a second foramen is found further anteriorly in Yuanmouraptor. This resembles that of Sinraptor (Currie & Zhao, 1993) and Concavenator (Cuesta et al., 2018). While in Allosaurus (Madsen, 1976a) and Acrocanthosaurus (Eddy & Clarke, 2011), there is only a single foramen presented.

The surangular forms most of the dorsal rim of the external mandibular fenestra. And an angular process of the surangular contributes to a slight part of the posteroventral rim of external mandibular fenestra. The angular process is evident in many allosauroids (Madsen, 1976a; Currie & Zhao, 1993; Gao, 1992; Gao, 1999; Eddy & Clarke, 2011), basal neotheropods (Rowe, 1989; Colbert, 1989), and abelisaurids (Sampson & Witmer, 2007; Bonaparte, Novas & Coria, 1990). Whereas this process is poorly developed in Mapusaurus (Coria & Currie, 2006). At the posteroventral corner of the external mandibular fenestra, the surangular is overlapped laterally by the posterodorsal border of the angular. Medially the surangular bears a hook-like process (Figs. 12E, 12F ), which forms an angle of nearly 60° with the long axis of the surangular and ventrally contacts the dorsal margin of the prearticular (Figs. 13C, 13D). The posterior rim of the hooked process extends laterally and forms the anterior wall of the glenoid. In medial view (Figs. 12E, 12F), the surangular thickens transversely to form a bar like dorsal rim, which was labeled as the medial shelf of the surangular by Eddy & Clarke (2011), demarcating the dorsal limit of the adductor fossa. The M. adductor mandibulae externus is estimated to attach the concave surface between the medial shelf and the lateral surangular ridge.

Figure 13 Mandibular joint of Yuanmouraptorjinshajiangensis gen. et sp. nov. (LFGT-ZLJ0115).

Right mandibular joint in (A) lateral view with (B) labeled drawing and in (C) dorsal view with (D) labeled drawing. Abbreviations: ar, articular; ct, foramen of chorda tympani; hp, hook like process of surangular; lg, lateral glenoid; mame, attachment of M. adductor mandibulae externus; mg, medial glenoid; par, prearticular; psf, posterior surangular foramen; r, ridge; retp, retroarticular process; sa, surangular; sr, surangular ridge. Scale bar represents 50 mm. Photos by Xiao-Chun Wu and Yi Zou.

Angular—The left angular lacks most of its anterior part (Figs. 12C, 12D), and the right angular is strongly dorsoventrally compressed. The angular forms the posteroventral part of the mandible, it thins dorsally to cover the surangular, and thickens ventrally to overlap the ventral margin of the prearticular, which forms the ventral border of the adductor fossa. The angular is dorsally concave, and forms the ventral rim of the external mandibular foramen. The preserved angular reaches the maximum depth at its central part where it contacts the surangular.

Prearticular—The right prearticular could be observed in medial view (Figs. 12E, 12F). The bone is a ventrally bowed element, it dorsoventrally flares at its anterior and posterior ends, but constricts in depth at its central part. The medial surface of the preserved prearticular is obscured by angular anteriorly and surangular posteriorly. The anterior part of the ventral rim of the prearticular is surrounded ventrally by the thickened angular. Whereas posterior to the contact with the angular, the ventral rim of the prearticular is exposed in lateral view as in Sinraptor (Currie & Zhao, 1993) and Acrocanthosaurus (Eddy & Clarke, 2011). The posterodorsal part of the prearticular forms a dorsomedially directed triangular process, which contacts the hooked process of surangular dorsolaterally (Figs. 13C, 13D). Posterior to this process, the prearticular forms a dorsally concave embayment, which houses the articular.

Articular—The right articular is well preserved and still in articulation with the prearticular and surangular (Fig. 13). The retroarticular process is well developed and bears a concave dorsal surface, which is dorsally oriented and bordered by a sharp ridge. In contrast, this surface faces more posteriorly in Allosaurus (Madsen, 1976a) and Acrocanthosaurus (Eddy & Clarke, 2011). In lateral view (Figs. 13A, 13B), the lateral rim of the retroarticular process surpasses the lateral wall of the surangular and is exposed laterally. In dorsal view, the posterior end of the retroarticular process is U-shaped in outline, then the process constricts its lateral rim abruptly and then expands medially at its anterior part, forming a lateral notch and a medial blunt process. This medial blunt process is contiguous with the medial rim of the medial glenoid, and the opening for the chorda tympani penetrates it from its posterodorsal surface to its ventral surface. Anteriorly, a prominent ridge separates the medial glenoid and retroarticular process as in Sinraptor (Currie & Zhao, 1993). This ridge is more pronounced in Acrocanthosaurus (Eddy & Clarke, 2011).

General description of the axial skeleton

Anterior 11 vertebrae are well preserved, with the missing of all ribs. The first 10 vertebrae were considered to represent the cervical series. The 11th might be the first dorsal vertebra based on the common condition of 10 cervical vertebra in basal-branching tetanurans (Holtz, Molnar & Currie, 2004) and the diapophysis of the eleventh vertebra which is more laterally expanded than the preceding one similar to Sinraptor (Currie & Zhao, 1993) and Yangchuanosaurus (Dong, Zhou & Zhang, 1983). The neurocentral suture is unobservable on the anterior and middle cervicals, and is only partly visible on the eighth and ninth cervicals, indicating that this individual is an adult or subadult.

Atlas-Axis—In the atlas-axial complex (Fig. 14), the atlantal intercentrum is in articulation with the axial intercentrum and odontoid (atlantal centrum). The proximal parts of both the left and right neurapophyses are still attached to the atlantal intercentrum and are positioned laterally to the neural canal. There is no evident prezygapophysis on the neurapophysis, indicating the absence of the proatlas as in Sinraptor (Currie & Zhao, 1993). The exposed part of the atlantal intercentrum is similar to that of Shidaisaurus (Wu et al., 2009) and Sinraptor (Currie & Zhao, 1993). In anterior view, the main body of the atlantal intercentrum is slightly wider than deep. The ventrolateral rim is slightly convex, and the ventral rim becomes flat, resulting a sub-rectangular profile of the lower half. The anterior articular surface is concave, for the articulation with the occipital condyle. The dorsal rim of the atlantal intercentrum is ventrally depressed and underlaps the rounded ventral part of the odontoid.

Figure 14 Atlas-axis complex of Yuanmouraptorjinshajiangensis gen. et sp. nov. (LFGT-ZLJ0115).

Atlas-axis complex in (A) dorsal, (B) ventral, (C) anterior, (D) posterior, (E) right lateral, (F) left lateral views, and labeled drawing of right lateral view (G). Abbreviations: ati, atlantal intercentrum; axi, axial intercentrum; dp, diapophysis; ep, epipophysis; fo, fossa; na, neurapophysis; nc, neural canal; ns, neural spine; od, odontoid; poz, postzygapophysis; pp, parapophysis; r, ridge; re, rounded eminence; spol, spinopostzygapophyseal lamina; tpol, intrapostzygapophyseal lamina. All names of the bony laminae follow terminologies in Wilson (1999). Scale bar represents 50 mm. Photos by Xiao-Chun Wu.

The odontoid (atlantal centrum) adheres to the upper half of the anterior articular surface of the axial centrum, and its sutures with the axial intercentrum and axial centrum are visible. The odontoid is divided into a dorsal and an anterior surface by an anteriorly convex rim, and this rim is approximately parallel with the ventral border which contacts the atlantal intercentrum. The dorsal surface of odontoid is dorsally concave and continuous with the floor of the neural canal. A pair of shallow recesses are located laterally on each side of the anterior surface of the odontoid. In contrast, there are a pair of foramina penetrating these recesses in Neovenator (Brusatte, Benson & Hutt, 2008). In anterior view, the odontoid is in shape of semicircle, similar to the condition in Allosaurus (Madsen, 1976a) and Monolophosaurus (Zhao et al., 2009).

The axial intercentrum is not observable in anterior view (Fig. 14C) due to the obscuration of the articulated atlantal intercentrum. In lateral view, the axial intercentrum tapers posterodorsally to form a sub-triangular outline (Fig. 14E) as in Neovenator (Brusatte, Benson & Hutt, 2008), Piatnitzkysaurus (Bonaparte, 1986), and Yunyangosaurus (Dai et al., 2020). In contrast, the axial intercentrum maintains a constant anteroposterior thickness dorsoventrally, resulting a sub-rectangular outline in lateral view in Sinraptor (Currie & Zhao, 1993), Yangchuanosaurus (Dong, Zhou & Zhang, 1983), and Ceratosaurus (Madsen & Welles, 2000). In lateral view, the suture between the axial intercentrum and centrum is slightly inclined anteroventrally, similar to the condition presented in Sinraptor (Currie & Zhao, 1993), but in contrast to the nearly vertical suture in Allosaurus (Madsen, 1976a), Piatnitzkysaurus (Bonaparte, 1986), Yunyangosaurus (Dai et al., 2020), and ceratosaurians (Madsen & Welles, 2000; Sampson & Witmer, 2007). The ventral surface of the axial intercentrum is flat and faces ventrally, being continuous with the ventral surface of the axial centrum in lateral view as in Ceratosaurus (Madsen & Welles, 2000), Dilophosaurus (Marsh & Rowe, 2020), Majungasaurus (O’Connor, 2007), Piatnitzkysaurus (Bonaparte, 1986), and Yunyangosaurus (Dai et al., 2020). This is different from Sinraptor (Currie & Zhao, 1993), Yangchuanosaurus (Dong, Zhou & Zhang, 1983), and Acrocanthosaurus (Harris, 1998), in which the ventral surface of the axial intercentrum faces more anteroventrally and is not continuous with the ventral surface of the centrum. In ventral view (Fig. 14B), the suture between the intercentrum and centrum is well exposed and arches anteriorly in Yuanmouraptor.

The axial centrum is proportionately longer compared to that of Ceratosaurus (Madsen & Welles, 2000), Eustreptospondylus (Sadleir, Barrett & Powell, 2008), and Yunyangosaurus (Dai et al., 2020). In lateral view, the axial centrum is longer ventrally than it is dorsally, as in Sinraptor (Currie & Zhao, 1993), Ceratosaurus (Madsen & Welles, 2000), and Piatnitzkysaurus (Bonaparte, 1986). This differs from Eustreptospondylus (Sadleir, Barrett & Powell, 2008) and Yunyangosaurus (Dai et al., 2020), in which the centrum is longer dorsally than ventrally. The anterior articular surface is mostly obscured by the odontoid and axial intercentrum. The posterior articular surface is strongly concave, and is approximately as wide as high, similar to Nevenator (Brusatte, Benson & Hutt, 2008) and Allosaurus (Madsen, 1976a), whereas the posterior articular surface is strongly mediolaterally compressed in Acrocanthosaurus (Harris, 1998). In addition, abelisaurids (Bonaparte, Novas & Coria, 1990; O’Connor, 2007) bear the posterior articular surface wider than long. In lateral view, the ventral rim of the centrum strongly arches dorsally, and forms an acute angle with the posterior articular surface as in Sinraptor (Currie & Zhao, 1993). The diapophysis is located at the base of the neural arch, just above the centrum, and is presented as a lateroventrally oriented pedicle. Anteroventral to the diapophysis, the parapophysis protrudes posteroventrally to form a diminutive hump. The lateral surface of centrum is smooth and there is no trace of any pneumatic structures. This resembles Piatnitzkysaurus (Bonaparte, 1986), Asfaltovenator (Rauhut & Pol, 2019), Shidaisaurus (Wu et al., 2009) and some individuals of Allosaurus (Benson et al., 2012), but contrasts with Sinraptor (Currie & Zhao, 1993), Neovenator (Brusatte, Benson & Hutt, 2008), and Acrocanthosaurus (Harris, 1998), in which the pneumatic foramen invades the main body of the centrum ventral to the diapophysis. The central part of the centrum tapers its transverse width downwards, resulting an mediolaterally narrow ventral surface with a weak midline ridge running the anterior half of the centrum. This differs from the centrum with mediolaterally wide and smooth ventral surface in Allosaurus (Madsen, 1976a), Sinraptor (Currie & Zhao, 1993), and Shaochilong (Hu, 1964; Brusatte et al., 2010b), in which no ridge is developed. A weak ventral ridge is also presented in Asfaltovenator (Rauhut & Pol, 2019), Eustreptospondylus (Sadleir, Barrett & Powell, 2008), Neovenator (Brusatte, Benson & Hutt, 2008), and abelisauroids (Carrano, Loewen & Sertich, 2011; O’Connor, 2007; Bonaparte, Novas & Coria, 1990). However, the axial centrum bears a pronounced ventral keel in Ceratosaurus (Madsen & Welles, 2000), basal neotheropods (Rowe, 1989; Marsh & Rowe, 2020), and Acrocanthosaurus (Harris, 1998). In Yunyangosaurus (Dai et al., 2020), the ventral surface of the centrum is transversely narrow as in Yuanmouraptor, but it is rounded and lacks a ventral ridge.

The prezygapophyses are obscured by the neurapophyses on both left and right sides (Figs. 14E, 14F). The postzygapophyseal facets of the axis are well developed, comparable in size to those of postaxial cervicals. The V-shaped intrapostzygapophyseal lamina (tpol) medially connects both postzygapophyses dorsal to the neural canal. The well-developed epipophysis extends posterodorsally to form a pointed tip and curves more laterally than the postzygapophysis, as in Shidaisaurus (Wu et al., 2009), Sinraptor (Currie & Zhao, 1993), and Neovenator (Brusatte, Benson & Hutt, 2008). Whereas in Allosaurus (Madsen, 1976a) and Monolophosaurus (Zhao et al., 2009), the epipophysis is low and its tip does not surpass the postzygapophysis laterally.

The neural spine is transversely thin and posterodorsally inclined. In lateral view, the neural spine deflects at an angle of 50° with the neural canal extended throughout its ventral half, then the degree of inclination decreases abruptly in the dorsal part. However, the degree of deflection is probably caused by breakage. Anterior to the base of the neural spine, a rounded eminence (Figs. 14A, 14E) rises dorsally, similar to the condition in Neovenator (Brusatte, Benson & Hutt, 2008) and Yunyangosaurus (Dai et al., 2020). The anterior end of the neural spine is positioned posterior to the anterior margin of the neural canal, contrasting the strongly anteriorly flared neural spine in Dilophosaurus (Marsh & Rowe, 2020) and Carnotaurus (Bonaparte, Novas & Coria, 1990). The spinopostzygapophyseal lamina (spol) expands laterally from the neural spine and fills the space between its summit and the epipophysis, resembling that of Sinraptor (Currie & Zhao, 1993; Gao, 1992; Gao, 1999), Yangchuanosaurus (Dong, Zhou & Zhang, 1983), Dilophosaurus (Marsh & Rowe, 2020), and Yunyangosaurus (Dai et al., 2020). In posterior view (Fig. 14D), an anteriorly excavated and subtriangular fossa is surrounded by the spinopostzygapophyseal laminae and postzygapophyses, such fossa also occurs in Sinraptor (Currie & Zhao, 1993; Gao, 1992; Gao, 1999), Yangchuanosaurus (Dong, Zhou & Zhang, 1983), Ceratosaurus (Madsen & Welles, 2000), and Dilophosaurus (Marsh & Rowe, 2020).

Figure 15 Postaxial vertebrae of Yuanmouraptorjinshajiangensis gen. et sp. nov. (LFGT-ZLJ0115).

Postaxial vertebrae in lateral view. (A) Cervical 3; (B) Cervical 4; (C) Cervical 5; (D) Cervical 6; (E) Cervical 7; (F) Cervical 8; (G) Cervical 9; (H) Cervical 10; (I) Dorsal 1. Abbreviations: acdl, anterior centrodiapophyseal lamina; adp, anterodorsal process; cpol, centropostzygapophyseal lamina; cprl, centroprezygapophyseal lamina; dp, diapophysis; ep, epipophysis; ns, neural spine; pcdl, posterior centrodiapophyseal lamina; pn, pneumatic foramen; podl, postzygadiapophyseal lamina; poz, postzygapophysis; pp, parapophysis; prdl, prezygadiapophyseal lamina; prz, prezygapophysis; spol, spinopostzygapophyseal lamina; ri, distinct rim on the anterior articular surface; sprl, spinoprezygadiapophyseal lamina; vk, ventral keel. All names of the bony laminae follow terminologies in Wilson (1999). Scale bar represents 50 mm. Photos by Xiao-Chun Wu.

Figure 16 Postaxial vertebrae of Yuanmouraptorjinshajiangensis gen. et sp. nov. (LFGT-ZLJ0115).

Cervical 3 in (A) anterior, (B) posterior, (C) ventral, (D) dorsal views. Cervical 5 in (E) ventral, (F) posterior views. Cervical 9 in (G) anterior, (H) posterior, (I) dorsal views. Abbreviations: cpol, centropostzygapophyseal lamina; cprl, centroprezygapophyseal lamina; ep, epipophysis; nc, neural canal; ns, neural spine; pn, pneumatic foramen; podl, postzygadiapophyseal lamina; poz, postzygapophysis; prz, prezygadiapophyseal lamina; ri, distinct rim on the anterior articular surface; tpol, intrapostzygapophyseal lamina; tprl, itraprezygapophyseal lamina; spol, spinopostzygapophyseal lamina; vk, ventral keel. All names of the bony laminae follow terminologies in Wilson (1999). Scale bar represents 50 mm. Photos by Xiao-Chun Wu.

Postaxial vertebrae—The third to fifth cervical centra are slightly opisthoceolous with dorsally arched ventral surface (Figs. 15A–15C). In subsequent elements of the cervical series, the anterior articular surface becomes flat or slightly convex, and their corresponding posterior articular surface becomes less concave, resulting the platycoelous centra. This is similar to the condition in Monolophosaurus (Zhao & Currie, 1993; Zhao et al., 2009), Majungasaurus (O’Connor, 2007), and Masiakasaurus (Carrano, Loewen & Sertich, 2011). The tendency of the reduction of the convexity and concavity of the anterior and posterior articular surface respectively also occurs in Sinraptor dongi (Currie & Zhao, 1993) and Sinraptor hepingensis (Gao, 1992; Gao, 1999), but all postaxial centra of these taxa are evidently opisthoceolous. The platycoelous condition is common in Ceratosaurus (Madsen & Welles, 2000), Dilophosaurus (Marsh & Rowe, 2020), and many early basal tetanurans (Bonaparte, 1986; Rauhut, 2005; Rauhut & Pol, 2019). A distinct rim (Figs. 16A–16C) is presented on the anterior articular surface of the anterior cervicals, and it is especially well defined on the third cervical. This distinct rim also occurs on anterior cervicals of Majungasaurus (O’Connor, 2007), Masiakasaurus (Carrano, Loewen & Sertich, 2011), Neovenator (Brusatte, Benson & Hutt, 2008), Baryonyx (Charig & Milner, 1997), and Yunyangosaurus (Dai et al., 2020), whereas all the postaxial cervical vertebrae bear such rim in Torvosaurus (Britt, 1991). In all postaxial cervicals, the anterior and posterior articular surfaces are slightly wider than high.

The centra of the postaxial cervicals are anteroposteriorly longer than dorsoventrally high, whereas the length/height ratio gradually decreases posteriorly through the cervical series. In comparison, the centrums are longer in proportion than those of Torvosaurus (Britt, 1991), Asfaltovenator (Rauhut & Pol, 2019), and Yunyangosaurus (Dai et al., 2020) but notably shorter than those of Coelurus (Carpenter et al., 2005), noasaurids (Carrano, Loewen & Sertich, 2011; Rauhut & Carrano, 2016), and basal neotheropods (Colbert, 1989; Rowe, 1989; Marsh & Rowe, 2020). In the third and fifth cervical vertebrae, the articular surfaces of the centrums are strongly offset from each other. Thus, the anterior articular surface is positioned more dorsally than the posterior articular surface, and the anterior articular surface is slightly ventrally oriented relative to the posterior articular surface. However, the anterior articular surface of the 10rd cervical vertebra is notably anterodorsally oriented. This inclination of articular surface indicates that the curved arrangement occurs in the neck of Yuanmouraptor as in most theropods except Carcharodontosaurus (Brusatte & Sereno, 2007), in which the offset between articular surface is absent.

The lateral surface of each of the postaxial cervical centra is excavated by a single pneumatic fossa, which is positioned posterodorsal to the parapophysis. In the third cervical (Fig. 15A), the pneumatic fossa deeply invades the ventrolateral part of the centrum, leading to a strongly transversely constricted ventral surface and a moderate ventral keel in the central part of the centrum (Fig. 16C). A sharp ridge runs laterally from the anterior end of the ventral keel towards the bottom of the parapophysis, similar to the condition in Dilophosaurus (Marsh & Rowe, 2020), but more pronounced than in Condorraptor (Rauhut, 2005). Posterior to the ventral keel, the ventral surface of the centrum expands transversely throughout the posterior part. In contrast, the third cervical of Eustreptospondylus (Sadleir, Barrett & Powell, 2008) lacks ventral keel. Although the third centrum bears a weak ventral keel in Dilophosaurus (Marsh & Rowe, 2020), Allosaurus (Madsen, 1976a), and Yunyangosaurus (Dai et al., 2020), the ventral surface of it is relatively broad. The ventral surfaces of the following cervical vertebrae become broad and flat without traces of ventral keels except the ninth and 10th cervical vertebra. In the fourth and fifth cervical, the pneumatic foramen is mediodorsally oriented and penetrates the floor of the neural canal. In the eighth and ninth cervical, the pneumatic foramen is anteroposteriorly elongated and dorsally roofed by a thin lamina. In the ninth cervical vertebra, a very weak ventral keel appears. In the 10th cervical vertebra, the ventral keel is enhanced with a prominent hypapophysis positioned anterior to the ventral keel, as in Sinraptor (Currie & Zhao, 1993), whereas the ventral surfaces of the third to 10th cervical vertebra lack midline ridges in Eustreptospondylus (Sadleir, Barrett & Powell, 2008), Neovenator (Brusatte, Benson & Hutt, 2008), and Monolophosaurus (Zhao et al., 2009).

The diapophysis of each postaxial cervical is lateroventrally extended, and bears a smooth articular surface with ellipse profile. The parapophysis is positioned ventral to the mid-height of the centrum and adjacent to the anterior articular surface in each cervical. In the third and the last two cervicals, the parapophysis is strongly shortened and ended with oval-shaped surface, whereas it protrudes lateroventrally in other cervicals. The laterally elongated parapophyses in medial cervicals also occur in Neovenator (Brusatte, Benson & Hutt, 2008). In the last cervical the parapophysis is posteriorly followed by a prominent ridge, which marks the bottom of the pneumatic foramen and shallows posteriorly.

The prezygapophysis projects anteriorly from the base of diapophysis, and is connected with the diapophysis laterally through the prezygadiapophyseal lamina (prdl). The paired prezygapophyses are well separated to be dorsolateral to the neural arch, and are medially connected by the intraprezygapophyseal lamina (tprl). The prezygapophysis has a sub-ellipse shaped and smooth articular facet, which faces anterodorsally and medially. The spinoprezygapophyseal lamina (sprl) laterally demarcates this facet, and connects the prezygapophysis with the neural spine. Throughout the whole cervical series this facet turns more anteromedially, and the anteroposterior distance between pre- and postzygapophysis gradually shortens. The posterodorsally and laterally projected postzygapophysis is more strongly developed than the prezygapophysis, with lateroventrally and posteriorly faced articular surface. In the last three cervicals (Figs. 15F–15H), the postzygapophyseal facets turn to face more posteriorly than preceding elements in the series. The postzygadiapophyseal lamina (podl) runs from the base of the diapophysis posteriorly to the distal end of the postzygapophysis or epipophysis. The spinopostzygapophyseal lamina (spol) originates posteriorly form the base of the neural spine and ends at the distal end of the articular surface of the postzygapophysis. Only the anterior three postaxial cervicals bear well-developed epipophysis, which emerges posterolaterally and dorsally to the postzygapophysis, similar to the posteriorly protruding condition in Concavenator (Cuesta, Ortega & Sanz, 2019). In the fourth and fifth cervicals, the epipophysis is extended well beyond the posterior margin of the postzygapophysis and forms a low plate-like structure, similar to but more prominent than that in Piatnitzkysaurus (Bonaparte, 1986). This differs from the more dorsally extended epipophysis in Sinraptor (Currie & Zhao, 1993), Torvosaurus (Britt, 1991), Eustreptospondylus (Sadleir, Barrett & Powell, 2008), and Ceratosaurus (Madsen & Welles, 2000). In addition, the epipophysis is extremely developed on the anterior cervicals in the former two taxa. In the subsequent cervical vertebrae of Yuanmouraptor, the epipophyses are absent and the dorsal surfaces of the postzygapophyses are flat and smooth. This contrasts to the condition in Allosaurus (Madsen, 1976a), Acrocanthosaurus (Harris, 1998), and Carnotaurus (Bonaparte, Novas & Coria, 1990), in which the epipophyses are retained in the whole cervical series.

The neural spines on the anterior cervical vertebrae are relatively low in comparison with those of Sinraptor (Currie & Zhao, 1993), Allosaurus (Madsen, 1976a), Acrocanthosaurus (Harris, 1998), but are similar in proportion to those of Monolophosaurus (Zhao et al., 2009) and Asfaltovenator (Rauhut & Pol, 2019). In the third and fourth cervicals (Figs. 15A, 15B), the anterior rim of the neural spine bears an anterodorsal process similar to the anterior shoulder in Dilophosaurus (Marsh & Rowe, 2020) and anterior projection in Baryonyx (Charig & Milner, 1997), resulting the abrupt discontinuity on the anterodorsal rim of the neural spine in lateral view. Such process might be for the attachment of ligament and also occurs in Acrocanthosaurus (Harris, 1998), despite in which this process protrudes more anteriorly and extends over the base of neural spine. The posterior rim of the neural spine in the third and fourth cervicals is posterodorsally convex, thus the posterior-most point is situated at nearly mid dorsoventrally height. In the subsequent cervicals, the neural spine increases in height progressively, and is dorsoventrally higher than anteroposteriorly long since the sixth cervical. In the eighth, ninth and tenth cervicals (Figs. 15F–15H), the neural spine is prominently dorsally elongated, with the distal end fanning out anteroposteriorly to become sheet-shaped in lateral view. Whereas in derived metriacanthosaurids (Dong, Zhou & Zhang, 1983; Currie & Zhao, 1993; Gao, 1992; Gao, 1999), the neural spines of the posterior cervical vertebrae are slender and rod-like. In the ninth (Figs. 16G–16I) and 10th cervicals, a shallow groove runs dorsoventrally along the anterior and posterior rims of the neural spine.

Several bony laminae connect the neural arch and the centrum and separate spaces between them into several pneumatic chambers as in most theropods. In the first five postaxial cervicals, the anterior centrodiapophyseal lamina (acdl) and centroprezygapophyseal lamina (cprl) are not very developed and laterally obscured by the ventrally oriented diapophyses. Due to the short distance between the neural arch and centrum in anterior cervicals, the centroprezygapophyseal laminae (cpol) of these elements are weakly developed. In the rest of the cervical series the distance between the arch and centrum increases with the elevation of the diapophysis from the parapophysis, resulting that aforementioned three bony laminae become more prominent. In the first three postaxial cervicals, the postzygadiapophyseal laminae (podl) do not continue onto the pedicle of the diapophyses. In the third cervical, this lamina is especially weakly developed and even discontinuous with the base of the postzygapophysis. In the last three cervicals, a pneumatic fossa excavates anteriorly into the ventral surface of the postzygadiapophyseal lamina (podl), as in Sinraptor (Currie & Zhao, 1993). All postaxial cervicals bear notable prezygadiapophyseal laminae (prdl) and posterior centrodiapophyseal laminae (pcdl).

The first dorsal vertebra (Fig. 15I) is platycoelous, with a flat anterior articular surface and a slightly concave posterior articular surface. The anterior and posterior articular surface are slightly wider than high as in cervical vertebrae. The neural arch and centrum of the first dorsal vertebra are in further distance compared to those in the cervical vertebrae. This extension is followed by the more elongated bony laminae connecting the neural arch and the centrum. The diapophysis is more laterally and horizontally oriented instead of pointing ventrolaterally. The parapophysis does not protrude laterally and its articular surface is immediately lateral to the anterior articular surface of the centrum and subtriangular in shape. The pre- and postzygapophysis decrease in height, followed by the reduction of inclination of the pre- and postzygadiapophyseal lamina in lateral view. The neural spine is approximately 1.5 times as dorsoventrally high as anteroposteriorly long, with a sub-rectangular lateral profile. The groove running along the anterior rim of the neural spine excavates deeper than those of former cervical vertebrae. As in postaxial cervicals, a single pneumatic foramen penetrates either side of the centrum, but these foramina on both left and right side are straightly connected without any bony walls. The ventral part of the centrum is similar to that of the last cervical vertebra, with a ridge originated from the posterior end of the parapophysis forming the lateral floor of the pneumatic foramen, and a developed ventral keel running through the ventral midline of the centrum.

Figure 17 Time-Calibrated strict consensus tree showing the phylogenetic position of Yuanmouraptor.

The most parsimonious trees were calibrated against geological time using the Rpackages Paleotree (Bapst, 2012) and Strap (Bell & Lloyd, 2015).

Phylogenetic analysis

The equally weighting phylogenetic analysis resulted in 1,152 MPTs, with each MPTs having a length of 1,282 steps (CI = 0.360, RI = 0.660). The strict consensus tree (Fig. 17) places Yuanmouraptor at the most ‘basal’ position in the Metriacanthosauridae, forming a polytomy with Xuanhanosaurus and the least inclusive group comprising Yangchuanosaurus and Sinraptor. In the phylogenetic result, the placement of Yuanmouraptor jinshajiangensis is supported by following characters: lacrimal horn developed as a small rugosity (character 50-0), shared with megalosauroids and basal neotheropods; postorbital bearing small anterior prominence (character 69-0), shared with megalosauroids and basal neotheropods; anterior articular surface of anterior cervical centra slightly convex (character 174-0), shared with early basal tetanurans such as Asfaltovenator and piatnitzkysaurids, as well as ceratosaurians; axis being in lack of pneumatic foramen (character 186-0), shared with piatnitzkysaurids and Coelophysis; perimeter of cervical anterior articular surface rimmed by a flattened peripheral band, forming a distinct rim (character 194-1), shared with megalosauroids; sheet-like and sub-rectangular posterior cervical neural spines (character 200-3); anterior dorsal vertebrae bears pronounced ventral keel (character 204-1), shared with Metriacanthosaurinae and Piatnitzkysaurus. The Metriacanthosauridae in our analysis is supported by seven synapomorphies: squamosal forms a flange covering quadrate head laterally (character 87-1); acute angle between the occipital condyle and the basal tubera (character 123-1); external mandibular fenestra is longer than 15% of the total mandible length (character 133-1); dorsoventral depth of surangular above external mandibular fenestra less than half of the height of mandible (character 134-0); well developed and broad spinopostzygapophyseal lamina (character 183-0); manus shorter than arm plus forearm (character 268-0); presence of metacarpal IV but lack of IV phalanges and whole digit V (character 269-2) (characters mapping see the online File S2 for details). Furthermore, three major branches of basally branching tetanurans (Megalosauroidea, Coeluerosauria, and Allosauroidea) are supported by this phylogenetic analysis. A monophyletic Carnosauria supported by Rauhut (2003) and Rauhut & Pol (2019) is not recovered, and Piatnitzkysauridae is placed as the sister-group to Avetheropoda (Allosauroidea + Coelurosauria).

In all results of implied weighting analyses (see the online File S3), Piatnitzkysauridae is stably placed as the sister group to Avetheropoda. And the internal toplogy of Metriacanthosauridae is generally identical to that of the equally weighting analyses when k = 6, k = 9, and k = 12. Yunyangosaurus becomes the sister taxon to Yuanmouraptor when k = 3 and k = 6, this alternative position of the former might because of its fragmentary nature. Thus, the discussion part will mainly focus on the result of equally weighting analysis.

Constraint analysis

Our phylogenetic analysis recovered Piatnitzkysauridae as the sister-group to Avetheropoda, this result is different from that it has closer relationship with Megalosauroidea supported by many previous researches (Benson, 2010; Carrano, Benson & Sampson, 2012; Rauhut, Hübner & Lanser, 2016; Dai et al., 2020; Rauhut et al., 2024). Also, the affiliation to Metriacanthosauridae of Xuanhanosaurus differs from the results of Benson (2010), Rauhut, Hübner & Lanser (2016), Rauhut & Pol (2019), Dai et al. (2020); and Rauhut et al. (2024), in which it was recovered as a member of Piatnitzkysauridae. To examine the stability of the phylogenetic position of Xuanhanosaurus and Piatnitzkysauridae in our result, we perform a constraint analysis to build three constraint phylogenetic topologies (Fig. 18) using TNT v. 1.6 (Goloboff & Morales, 2023). In the first topology, Piatnitzkysauridae is constrained to be part of Megalosauroidea as the sister group of all other megalosauroids. In the second topology, the Xuanhanosaurus is constrained to belong to Piatnitzkysauridae. In the third topology, the Xuanhanosaurus is constrained to belong to Piatnitzkysauridae, which is constrained to be part of Megalosauroidea as the sister group of all other megalosauroids. The settings of phylogenetic analysis are identical to that of the unconstrained analysis. The result is shown in the Table 1, and cladograms are shown in the online File S4.

Figure 18 Alternative phylogenetic topologies for constraint analysis.

(A) Piatnitzkysauridae is placed as a basal branch within Megalosauroidea. (B) Xuanhanosaurus is placed as a basal member of Piatnitzkysauridae. (C) Xuanhanosaurus is placed as a basal member of Piatnitzkysauridae with Piatnitzkysauridae placed as a basal branch within Megalosauroidea.

Table 1 Results of phylogenetic analysis with Piatnitzkysauridae and Xuanhanosaurus constrained at different positions.

Topology	Unconstrained	Piatnitzkysauridae as a sister group to the clade comprising Megalosauridae and Spinosauridae	Xuanhanosaurus as a member of Piatnitzkysauridae	Piatnitzkysauridae as a sister group to the clade comprising Megalosauridae and Spinosauridae, and Xuanhanosaurus as a member of Piatnitzkysauridae	
Tree structure changes		Members of Piatnitzkysauridae fall into a polytomy; Streptospondylus, Eustrptosponylus, and a monophyletic group comprising other megalosaurids fall into a polytomy at the base of Megalosauridae	Yuanmouraptor, ‘Szechuanosaurus’ zigongensis, Shidaisaurus, (CNM V214+Yangchuanosaurus), and Metriacanthosaurinae in polytomy at base of Metriacanthosauridae	Yuanmouraptor, ‘Szechuanosaurus’ zigongensis, Shidaisaurus, (CNM V214+Yangchuanosaurus), and Metriacanthosaurinae in polytomy at base of Metriacanthosauridae	
TL	1,282	1,284	1,284	1,286	
MPT	1,152	2,304	2,304	2,304	
CI	0.360	0.360	0.360	0.359	
RI	0.660	0.660	0.660	0.659	
Synapomorphies of Piatnitzkysauridae	23(0), 184(1), 185(1), 186(0), 209(2), 256(1)	23(0), 157(1), 184(1), 185(1), 186(0), 212(1), 216(1), 256(1)	245(1)	245(1), 257(1)	
Synapomorphies of Metriacanthosauridae	87(1), 123(1), 133(1), 134(0), 183(0), 268(0), 269(2)	87(1), 123(1), 133(1), 134(0), 183(0), 268(0), 269(2)	71(0), 87(1), 123(1), 133(1), 134(0), 181(1), 183(0), 227(1), 252(0), 318(1), 319(2), 320(1)	71(0), 87(1), 123(1), 133(1), 134(0), 181(1), 183(0), 227(1), 252(0), 318(1), 319(2), 320(1)	
Notes.

Abbreviations CI consistency index

MPT most parsimonious tree

RI retention index

TL tree length

Format of the table follows that of Nesbitt et al. (2014).

Discussion

Comparison with Jurassic metriacanthosaurids in China and morphology modification within Metriacanthosauridae

As shown in the result of our phylogenetic analysis (Fig. 17), Yuanmouraptor and Xuanhanosaurus are placed outside the monophyletic group, which includes Metriacanthosaurinae (Carrano, Benson & Sampson, 2012), Yangchuanosaurus, and their latest ancestor and all descendants. And this results a basal trichotomy within Metriacanthosauridae.

Sinraptor dongi (Currie & Zhao, 1993), S. hepingensis (Gao, 1992; Gao, 1999), and Yangchuanosaurus shangyouensis (Dong et al., 1978; Dong, Zhou & Zhang, 1983) represent derived members of Metriacanthosauridea and all lived in the Late Jurassic. Materials of these taxa reach a high degree of completeness, and provide significant taxonomic information. These three taxa are large-sized theropods, and the skull length could reach approximately two times the condition in Yuanmouraptor. In S. dongi, S. hepingensis, and Y. shangyouensis, the skull and vertebrae are highly pneumatized, such as the pneumatic foramen on the lateral surface of the jugal and flank of the axial centrum, contrasting the poorly-developed pneumatic structure in Yuanmouraptor. The ventral process of the postorbital in three derived metriacanthosaurids has a small suborbital flange, which might mark the ventral limit of the eyeball, whereas in Yuanmouraptor the ventral process of the postorbital is smooth and slightly concave below the eyeball. All three derived metriacanthosaurids bear rugose ornaments on the upper part of the skull, such as heavy rugosity on the nasals, well-developed lacrimal horns, and a large rugose boss forming the anterior process of the postorbital. Though the nasal of Yuanmouraptor was not preserved, the lacrimal and postorbital only possess slight rugosity. The cervical vertebrae of derived metriacanthosaurids are strongly opisthocoelous with ball and socket articular surface, and the axial intercentrum is anterodorsally flexed. This condition shows greater mobility (Snively et al., 2013) than that of Yuanmourpator, in which the cervical vertebrae are platycoelous and the ventral surface of the axial intercentrum is continuous with that of the centrum. The anteroposteriorly narrow neural spines in posterior cervical vertebrae in derived metriacanthosaurids are also different from the sheet-like neural spines of posterior cervical vertebrae in Yuanmouraptor. However, whether this distinction of morphology in neural spine leads to the difference of neck mobility is not clear.

Because there are alternative phylogenetic relationships within Metriacanthosauridae (Rauhut, Hübner & Lanser, 2016; Rauhut & Pol, 2019; Rauhut et al., 2024), here we do not define any taxonomic groups within Metriacanthosauridae. Within the monophyletic clade excluding Yuanmouraptor and Xuanhanosaurus in the Metriacanthosauridae, one branch comprises all metriacanthosaurids more closely related to Yangchuanosaurus (Dong et al., 1978; Dong, Zhou & Zhang, 1983) than to Metriacanthosaurus, and another branch was defined as Metriacanthosaurinae by Carrano, Benson & Sampson (2012). These two branches are distinct in many aspects. In the axial skeleton, the anterior dorsal vertebrae of Metriacanthosaurinae have prominent ventral keels, whereas in another branch the keel is weakly developed. In the pelvic girdle, the angle between the long axis of pubis and pubic boot is less than 60° in Metriacanthosaurinae, but nearly perpendicular in the branch represented by Yangchuanosaurus. In Metriacanthosaurinae the ischial shaft is ventrally curved and the distal end is slightly expanded, while the ischial shaft is straight and the distal end of ischium is notably expanded to form an ischial boot in Yangchuanosaurus branch. The slightly expanded distal end of the ischium in Y. shangyouensis and Metriacanthosaurinae might suggests that this trait is gained independently in these taxa. In the hindlimbs, members of Metriacanthosaurinae possess bulbous fibular crest on the tibia, in contrast to the narrow and lamina-like fibular crest of the tibia in the branch represented by Yangchuanosaurus.

The Middle Jurassic Xuanhanosaurus (Dong, 1984) was considered to belong to Megalosauroidea as a member of Piatnitzkysauridae in previous studies (Benson, 2010; Rauhut, Hübner & Lanser, 2016; Rauhut & Pol, 2019; Dai et al., 2020; Rauhut et al., 2024). In our phylogenetic analysis, Xuanhanosaurus is recovered as a basal member of Metriacanthosauridae, as in Carrano, Benson & Sampson (2012), but this placement is poorly supported with only two characters: relatively short manus, and developed metacarpal IV with lack of IV phalanges and digit V, shared with CNM V214 (Dong, Zhou & Zhang, 1983, the specimen number follows Wu et al., 2009) and ‘Szechuanosaurus’ zigongensis (Gao, 1993) respectively. In the constraint analysis, we constrain Xuanhanosaurus as a piatnitzkysaurid in the second constraint topology, and then constrain it as a piatnitzkysaurid with the Piatnitzkysauridae belonging to Megalosauroidea in the third constraint topology. The result of the constraint analysis shows that placing Xuanhanosaurus in Piatnitzkysauridae needs extra two steps. Based on that, if additionally force Piatnitzkysauridae into Megalosauroidea, four extra steps are needed. Whether in the second or the third topology, the placement of Xuanhanosaurus from Metriacanthosauridae to Piatnitzkysauridae will erode the number of synapomorphies of Piatnitzkysauridae to only one or two. Removing Xuanhanosaurus from Metriacanthosauridae leads to the collapse of the branch comprising all metriacanthosaurids more closely related to Yangchuanosaurus than to Metriacanthosaurus, leaving that Shidaisaurus and ‘Szechuanosaurus’ zigongensis fall into a polytomy, also accompanied with the increase of synapomorphies count of Metriacanthosauridae. Thus, removing Xuanhanosaurus from Metriacanthosauridae to Piatnitzkysauridae lowers the stability of Piatnitzkysauridae and breaks the two main branches interrelationship in Metriacanthosauridae. On the other hand, to alter the position of Xuanhanosaurus only costs extra two or four steps in our phylogenetic analysis. In that case, phylogenetic position of Xuanhanosaurus still needs to be testified by a detailed study in the future. The overlapping materials of Xuanhanosaurus with Yuanmouraptor are limited to two posterior cervical vertebrae. The eighth cervical centrum of Xuanhanosaurus is evidently opisthocoelous, in contrast with the platycoelous condition in that of Yuanmouraptor.

Shidaisaurus (Wu et al., 2009) was found in Chuanjie Formation (Middle Jurassic) in Lufeng City, Yunnan, which is 122 km from the type locality of Yuanmouraptor. Shidaisaurus was the first tetanuran reported in Yunnan Province. The skull roof, dorsal part of the occiput, and axis of Shidaisaurus are preserved, and these elements could be compared with Yuanmouraptor. Both Yuanmouraptor and Shidaisaurus possess paired frontals broader than long, which generally occur in Allosauroidea; In contrast, in Megalosauroidea (Bonaparte, 1986; Allain, 2002; Li et al., 2009) and more basal therapods (Colbert, 1989; Rowe, 1989), the paired frontals are anteroposteriorly longer than transversely wide. Similar to Shidaisaurus, a slight margin of the frontal contributes to the dorsal margin of the orbit in Yuanmouraptor, this also occurs in other metriacanthosaurids (Dong, Zhou & Zhang, 1983; Gao, 1992; Gao, 1999; Currie & Zhao, 1993) and Allosaurus (Madsen, 1976a). The occiput of Shidaisaurus and Yuanmouraptor share a similar posteroventrally directed paroccipital process, with the ventral base located beneath the occipital condyle, a condition commonly developed in Allosauroidea. However, the supraoccipital of Yuanmouraptor forms a moderate part of the dorsal rim of the foramen magnum, differing from Shidaisaurus, in which the supraoccipital does not contribute to form the dorsal margin of the foramen magnum. An atlas-axis complex is the only preserved cervical element of Shidaisaurus. The axial centrum of Shidaisaurus bears some similarities to that of Yuanmouraptor, including broad spinopostzygapophyseal laminae and absent pneumatic foramina. The axial intercentrum of Shidaisaurus has an anterodorsal inclined ventral surface, followed by an anterodorsally faced anterior articular surface, which resembles the condition in the derived metriacanthosaurids Yangchuanosaurus (Dong, Zhou & Zhang, 1983) and Sinraptor (Currie & Zhao, 1993). However, the alignment of the atlas-axis complex is different in Yuanmouraptor, in which the ventral surface of the axial intercentrum is parallel with the axial centrum, but the anterior articular surface of the axial intercentrum faces anterodorsally, resulting a triangular lateral profile of the axial intercentrum. This might represent an early form of the arrangement of the atlas-axis complex as in many early basal tetanurans (Bonaparte, 1986; Zhao et al., 2009). Although Yuanmouraptor and Shidaisaurus share similar geological distribution and approximately contemporaneous stratigraphic unit which were known as Middle Jurassic (Huang, 2005; Fang et al., 2008), the difference in morphology along with the results of our phylogenetic analysis support the validity of Yuanmouraptor jinshajiangensis gen. et sp. nov.

The Late Jurassic CNM V214 (Dong, Zhou & Zhang, 1983) and the Middle Jurassic ‘Szechuanosaurus’ zigongensis (Gao, 1992) are also positioned as derived metriacanthosaurids by the phylogenetic analysis and are placed within the branch represented by Yangchuanosaurus. The first reports of CNM V214 and ‘S.’ zigongensis regarded them as the neotype of ‘Szechuanosaurus’ campi (Young, 1942) and a new species of the genus ‘Szechuanosaurus’ respectively. The type species of genus ‘Szechuanosaurus’, ‘S. campi’, was based on four isolated teeth, and was considered as invalid (Wu et al., 2009; Carrano, Benson & Sampson, 2012). Due to the lack of detailed restudies and phylogenetic analyses of these two specimens for decades, CNM V214 and ‘S.’ zigongensis have not been given formal taxonomic names so far. The information about CNM V214 is very limited, with part of the cervical series overlapping with Yuanmouraptor. The axial complex of CNM V214 is similar to that of S. dongi, S. hepingensis, and Yangchuanosaurus instead of Yuanmouraptor, with the anterodorsally tilted ventral surface of the axial intercentrum and well-developed pneumatic foramens on the centrum. The maxilla of ‘S.’ zigongensis is similar to that of Yuanmouraptor, with well-developed antorbital fossa. However, the morphology of the posterior cervical vertebrae of ‘S.’ zigongensis resembles the condition in those Late Jurassic taxa, in which the neural spines are anteroposteriorly narrow and rod-like.

Many character modifications occurred in Metriacanthosauridae from the Middle Jurassic to the Late Jurassic. First, as shown in Yuanmouraptor, the basal-branching Middle Jurassic members of this clade do not possess a well-developed pneumatic system as in those Late Jurassic descendants, such as the lack of pneumatic foramen on jugal and pneumatic foramina on axial centrum. The condition of lacking pneumatic structures also occurs in Shidaisaurus (Wu et al., 2009). Second, the ornamentations of the skull have been changed from a slight rugose brow in Yuanmouraptor to a prominent lacrimal horn and heavy rugosity on postorbital in Sinraptor (Gao, 1992; Currie & Zhao, 1993) and Yangchuanosaurus (Dong, Zhou & Zhang, 1983). Third, the alignment of the atlas-axis complex and morphology of cervical vertebrae have been changed to improve the mobility of the neck. In the basal form, as shown in Yuanmouraptor, the ventral surface of the axial intercentrum and centrum are continuous, and subsequent cervical vertebrae are platycoelous. This condition has been modified to that the ventral surface of axial intercentrum is notably inclined anterodorsally to bring the neck underneath the skull (Currie & Zhao, 1993) and cervical vertebrae are strongly opisthocoelous with ball-and-socket articular surface in Yangchuanosaurus and Sinraptor. Furthermore, the neural spines of posterior cervical vertebrae are sheet-like in Yuamouraptor but rod-like in those Late Jurassic metriacanthosaurids. Two Middle Jurassic taxa, Shidaisaurus and ‘Szechuanosaurus’ zigongensis (Gao, 1993) bear mosaic combination of characters. In ‘Szechuanosaurus’ zigongensis (Gao, 1993), the neural spines of posterior cervical vertebrae are also anteroposteriorly constricted and nearly rod-like, but the articular surfaces of cervical centra are platycoelous. Shidaisaurus (Wu et al., 2009) possesses anterodorsally inclined axial intercentrum, but lacks pneumatic foramnia on the axial centrum.

Implications on phylogeny of basal-branching tetanurans

Since three major branches (Megalosauroidea, Allosauroidea, and Coelurosauria) within Tetanurae were proposed by Carrano, Benson & Sampson (2012), alternative opinions (Rauhut & Pol, 2019; Lamanna et al., 2020; Schade et al., 2023; Rauhut et al., 2024) upon the interrelationship of tetanurans were put forward in past decade. The result recovered by our phylogenetic analysis also approaches to the three-major-clade pattern within Tetanurae with relatively high resolution. Among the nodes within Tetanurae, Spinosauridae and Coelurosauria (with the exception of Lourinhanosaurus) are well-supported, with Bremer support scored 3. Besides, Allosauria (Allosauridea + Carcharodontosauria) and Metriacanthosaurinae (Carrano, Benson & Sampson, 2012) are supported with Bremer support scored 2 (see the online File S5 for details). The low bootstrap support of many nodes within Megalosauridae might suggest their poorly-preserved condition (see the online File S6 for details). According to our constraint analysis, the placement of Piatnitzkysauridae close to Avetheropoda is not stable, with two additional steps needed to recover its affinity to Megalosauroidea (Table 1). Such placement of Piatnitzkysauridae within Megalosauroidea increases the number of synapomorphies supporting Piatnitzkysauridae, with the positions of Streptospondylus and Eustreptospondylus falling into a polytomy at the base of Megalosauridae. Besides, there are not any notable changes of phylogenetic relationship compared to the result of unconstraint analysis. Hence the phylogenetic position of Piatnitzkysauridae needs more detailed study to examine.

At least eight taxa or specimens from western China are positioned within the group Allosauroidea by our phylogenetic analysis. The finding of Yuanmouraptor provides an example of early stage in tetanuran evolution. Many characters presented in Yuanmouraptor are shared with megalosaurid theropods or non-tetanuran theropods, such as pneumatic foramen absent on jugal as in basal neotheropods; pneumatic foramen absent on the axial centrum as in basal neotheropods and Piatnitzkysaurus; platycoelous cervical centrum as in basal neotheropods, ceratosaurians, and piatnitzkysaurids; and the distinct rim on the anterior articular surface of cervical centrum (shared with Torvosaurus, Yunyangosaurus, Baryonyx, Majungasaurus, and Masiakasaurus). This indicates that high level of homoplasy among these Early and Middle Jurassic theropods. Thus, findings of key taxa to bridge the gap between non-tetanuran ancestors and a variety of derived tetanuran clades are important to testify whether similar character states in different clades are the result of homology or homoplasy. Meanwhile, the construction and sampling of characters, accuracy of state scores, and issues of sampling are also strongly related to alleviate phylogenetic uncertainty (Lovegrove, Upchurch & Barrett, 2024). The review and redescription of the named taxa or specimens are of great significance. There are still many taxa or specimens (Dong et al., 1978; He, 1984; Dong, Zhou & Zhang, 1983; Dong & Tang, 1985; Gao, 1992; Gao, 1993; Gao, 1999; Li et al., 2009) reported in China lacking detailed osteological descriptions. New anatomic information helping to resolve the phylogenetic problems will be extracted after the detailed re-examination and description of those taxa in future works.

Conclusions

A new metriacanthosaurid, Yuanmouraptor jinshajiangensis gen. et sp. nov, is established based on a relatively complete cranium, a complete cervical series without any ribs, and an anterior-most dorsal vertebra. Yuanmouraptor is diagnosed by a unique combination of characters, especially five autapomorphies. Phylogenetic analysis places Yuanmouraptor at a basal-branching position within Metriancanthosauridae. Yuanmouraptor presents the most complete cranium among basal-branching tetanurans reported in Middle Jurassic China, and provides valuable anatomic information concerning the unusual combination of plesiomorphies and synapomorphies of cranium and cervical vertebrae in Metriacanthosauridae. In addition, our phylogenetic analysis recovered the phylogenetic position of Piatnitzkysauridae being the sister group to Avetheropoda instead of being within Megalosauroidea. Three major branches within Tetanurae are recovered by our phylogenetic analysis with support of the monophyletic Avetheropoda (Allosauroidea + Coelurosauria) instead of the monophyletic Carnosauria (Megalosauroidea + Allosauroidea) proved by Rauhut (2003) and Rauhut & Pol (2019). Due to the lack of consensus upon the phylogenetic relationship within basal-branching tetanurans over past decades and many relatively fragmentary materials within Tetanurae, more accuracy in character coding and new findings of early members of this clade are needed to untangle the interrelationship of basal members of the group in the future.

Supplemental Information

Supplemental Information 1 Characters list

Supplemental Information 2 Characters mapping of the equally weighting strict consensus tree

Supplemental Information 3 Results of implied weighting analyses

Supplemental Information 4 Cladegrams of constraint analysis

Supplemental Information 5 Bremer supports

Supplemental Information 6 Bootstrap analysis

Supplemental Information 7 The matrix used for the phylogenetic analysis in our work

We are grateful to Lufeng World Dinosaur Valley Museum staff for giving the access to the fossil material of LFGT-ZLJ0115. We would like to thank Qian-Nan Zhang, Ya-Ming Wang, and Yan-Chao Wang, because this research greatly benefited from discussions with them. We appreciate the great effort made by responsible reviewers and editor to improve the manuscript and promote the review process.

Institutional Abbreviations

CNM Chongqing Natural History Museum

LFGT The Bureau of Natural Resources of Lufeng City, Yunnan, China.

Additional Information and Declarations

Competing Interests

Author Contributions

Data Availability

New Species Registration

Wei-Gang Zhang is employed by Chuxiong Jurassic Cultural Tourism Industrial Park Development Co. Ltd.

Yi Zou conceived and designed the experiments, performed the experiments, analyzed the data, prepared figures and/or tables, authored or reviewed drafts of the article, and approved the final draft.

Li Chen performed the experiments, prepared figures and/or tables, and approved the final draft.

Tao Wang performed the experiments, prepared figures and/or tables, and approved the final draft.

Guo-Fu Wang performed the experiments, prepared figures and/or tables, and approved the final draft.

Wei-Gang Zhang analyzed the data, prepared figures and/or tables, and approved the final draft.

Xiao-Qin Zhang analyzed the data, prepared figures and/or tables, and approved the final draft.

Zhen-Ji Wang analyzed the data, prepared figures and/or tables, and approved the final draft.

Xiao-Chun Wu performed the experiments, prepared figures and/or tables, authored or reviewed drafts of the article, and approved the final draft.

Hai-Lu You conceived and designed the experiments, authored or reviewed drafts of the article, and approved the final draft.

The following information was supplied regarding data availability:

The raw data is available in the Supplementary File.

The following information was supplied regarding the registration of a newly described species:

Genus name: Yuanmouraptor gen. nov. urn:lsid:zoobank.org:act:8F99DC0B-5E55-42CD-A0CF-9216F9EBE268.

Species name: jinshajiangensis gen. et sp. nov. urn:lsid:zoobank.org:act:5AE0D7CB-C337-41A2-BDC8-1F2E500624F6.

Publication LSID: urn:lsid:zoobank.org:pub:2A9F32AD-B671-4F48-8A6E- 0A69976A75FB.

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
