# Peer review of "A new metriacanthosaurid theropod dinosaur from the Middle Jurassic of Yunnan Province, China"

_PeerJ, doi:10.7717/peerj.19218_

## Round 0.1 · original submission · Major Revisions

· Academic Editor

Major Revisions

In particular, provide a stronger and more justified diagnosis of the new taxon and consider adding additional analyses (e.g., equally weighted and implied weighted) to further strengthen the phylogenetic investigation. In addition, a geological section would be a benefit to the reader.

·

Basic reporting

The manuscript "A new metriacanthosaurid theropod dinosaur from the Middle Jurassic of Yunnan Province, China" by Zou and colleagues describes new theropod material from China adding to the diversity of metriacanthosaurids.

The material itself is quite informative including skull material and the anterior part of the axial skeleton, thus, allowing to establish a new taxon with valid diagnostic characters. The material is well documented with multiple figures.

The general desciption of the material is generally okay, although the quality of the skull description is way better than that of the axial skeleton. The latter requires more work to revise. I am not a native speaker, but I tried to fix some of the phrases. Nevertheless, I would recommand to use shorter sentences if possible.

I also miss a little bit of geological context. The systematic section includes some basic information on the locality and the geologica formation, but I would encourage the authors to add a whole paragraph on the geological setting and figure showing the stratigraphy of the formation and in which part of the formation the fossil was found.

The performance of the phylogenetic analysis fulfills the standard and the results look sound to me. The long ghost-lineage of spinosaurids at the base of megalosauroids looks odd, but as this not the focus of the paper, I do not insist any revision. Nevertheless, the authors use the term Carnosauria wrong, both in the text and the illustration of the phylogeny. Carnosauria include Allosauroidea and Megalosauroidea, but not Coelurosauria. As coelurosaurs are the sister group of allosauroids in the present analysis, Carnosauria was not recovered as clade. This need to be changed.

Otherwise, I am happy with the mansucript and would support the publication after careful revision. I added a PDF with some notes and comments that need to be addressed.

All best,
Christian Foth

Experimental design

no comment

Validity of the findings

no comment

Additional comments

no comment

·

Basic reporting

Intro and background:

The Intro and Stage of the art do not reflect the significance of this discovery in time and paleobiogeography. This was better expressed in the abstract: “…the Late Jurassic is well represented by the nearly complete specimens, but the incompleteness of Middle Jurassic taxa hinders our knowledge of the origin and early evolution of Metriacanthosauridae”. After reading that in the abstract, I expected a wider explanation of the record of this clade in China, and in Asia, in general. A broader view of the knowledge about this clade would help understand the current and the whole context of Methiacanthosaurids. Also, relative to the phylogenetic context, I suggest a more comprehensive introduction of the nowadays possible discussion in early-branching tetanurans, concerning the phylogenetic position of metriacanthosaurid, to give the audience a better context (for example, comparing the most recent studies as Carrano et al., 2012, Wang et al., 2017; Rauhut and Pol, 2019, or Rauhut et al., 2024). Also, I have missed a robust Research Question, and the authors might remark on the importance of this material in all the contexts (paleobiogeographic, phylogenetic) in the Paleobiology of theropods and, historically, compared with the poorly described or poorly accessible other materials from China.

Comment on language and grammar issues:

In general, English is acceptable, but I suggest a double-check of the text because there are many grammar mistakes and misspellings. I have marked some of them here, but I have probably lost many others; please check it well.
-There are several highly long sentences with subordinate clauses, making the reading more difficult. For example: Line 143 to Line 151; Line 286 to Line 289; Line 370 to 372.
-Line. 47: misspelling “Material & Mathods”, for “Material & Methods”
-Line 133: Eliminated an extra space
-Line 143: I suggested eliminating the “outline of the” and leaving only the premaxillary body because it describes the morphology of the whole structure; it is not necessary to specify the “outline”.
-Line 172: Change “without the dorsal inclination” to “posteriorly oriented”
-Line 218: Eocarcharia in Italic: Eocarcharia
-Line 271: change “expends” to “expands”
-Line 271: delete “rest”
-Line 272: change “impossible” to “not possible”
-Line 273: change “disconnected” to “disarticulated”
-Line 286: change ·which lengthened” by “with a length of…”
-Line 300: “The anterodorsal border of jugal are broken on both left and right sides” change to “The anterodorsal border in the left and right jugals are broken”
-Line 302: change “remnant” to “preserved” (to apply in all of them)
-Line 324: Monolophosaurus in Italic: Monolophosaurus
-Line 345: Change “latter´s missing” to “the missing of the dorsal tip”
-Line 363: eliminated “as”
-Line 400: change “frontal” to “prefrontal”
-Line 665: “dorsoventrally deeper” change to “dorsoventrally deeper excavated” because it talks about a notch.
-Line 998: Change “Sinrptor” to “Sinraptor”
-Line 1074: change “craniums” to “cranium” (it is just one skull)

Comments on Figures:

General: I miss a figure with a skeletal drawing and the bones preserved in this specimen. It is usually helpful to understand how complete the specimen is and check possible overlapping of bones with new specimens or taxa that could be used for comparison.
Fig. 4: The scale bars in D are the same size, so using just one is enough. My suggestion is to avoid using the labels over the bones/teeth for as long as possible and the scale bars so close to the denticles. I suggested just one scale bar per figure in the corner of the picture and not over any fossil or information. Figure 4C is under focus in the second premaxillary tooth. Also, this picture aims to show the mesial and distal carina, but they are incomplete in the picture. So, I suggest a better photo showing the whole carina of both teeth to support the information in the text.

Fig. 10: I recommend including all the possible views for each bone in the exact figure. For example, the right surangular is also visible in the lateral view, but only the medial view is on the figure. To avoid checking Figure 2, it is easier for the lectures to see that the bones are fully and well-illustrated in their respective figures.
Fig. 12: Please, could you add a drawing of the Atlas-Axis to understand better the interpretations? Also, it is necessary to have an extra label on some anatomical elements that are mentioned in the text (e.g., the rounded eminence of the axial neural spine).
Fig. 13: Please, label the anterodorsal oriented process in the anterior rim of the neural spine.
Fig. 15: Remove the node Carnosauria label (see comments on the text). Change “Charcharodontosauria” by “Carcharodontosauria”.



RAW data:

Following the principles of FAIR, the data for phylogenetic analysis should be the nexus file or the character matrix uploaded in an available, accessible, free, and accurate repository such as Morphobank or in the supplementary of the paper, in addition to the TNT file. In a TNT file, the reuse of the data is only possible in TNT and not in other software, such as PAUP, for example. Also, checking the codification is not easy with the TNT file or it is not possible to modify to reuse the analysis with extra data. Therefore, the RAW data should be always the matrix in a modifiable file. Moreover, the TNT metadata is not enough to reproduce, replicate, or reuse the analysis (for example, name of the characters, info about taxa, etc.…). Still, it is possible to include this info in a Nexus file. I ask the author to include the nexus file or upload it in a valid repository that follows the FAIR principles for Open Access and include in it all the metadata needed to be properly reused and replicable.

Experimental design

Methodology:

I suggest improving the phylogenetic methodology. I explain better in the PDF in every part that I discuss, but in sum, I propose to do several additional analyses and changes in the methodology as: (1) order some characters; (2) equally weighting and Implied weighting analysis (following the recent recommendation by Ezcurra, 2024); (3) Constrain analyses of the alternative position of some taxa; (4) mapping of the character that support its placement in metriacanthosaurids and also the proposed synapomorphies for metriacanthosaurids using Mesquite or TNT. Include all the outcomes that are not included as figures in the supplementary data and all the raw materials that these analyses will generate.

Validity of the findings

Results:

The taxon's diagnosis is not strong enough. Many of the features labelled as autapomorphies are also observed in other metriacanthosaurids, and some of them and other unique combinations of features are doubtful or maybe preservation artifacts. Please look carefully at the comments throughout the PDF on this, check it, and provide better justifications and a stronger diagnosis.
In general, the description is good and with a strong comparison with other taxa (although It can be improved in the axial skeleton part) , not just allosauroids, but all theropods in general, and I appreciate so much a good description of new taxa, especially in this group, because most of the metriacanthosaurid from China are poorly described or in Chinese, so the information is reduced or inaccessible. However, I find the proper description short compared to the comparison part with other theropods information. So, I would like to suggest a better trade-off for the information. Because this group is historically poorly described, it will be very useful that this new specimen that preserves so many elements and is in good condition was deeply and widely described (see some examples in questions in the PDF of the main text). There are so many features that are described briefly, and in the picture, it is not possible to see very well, so if they were better and more detailed described, this would compensate for the potential lack of information in some photos. Also, do not extend the description section so much and to be easier for readers, I also suggest separating anatomical results into “description” (proper description of the fossils, without comparison) and “comparison” sections (where you find just the current comparison, that I think that are very useful and are complete as they are). But the latter is just a suggestion; this section could be as it is (comparing bone by bone), but please extend the proper description of the fossils.

Discussion:

I suggest including a section about the discussion of the characters that support the placement of Yuanmouraptor in this clade and discussing this character with their wider distribution among theropods. Therefore, it should be mapped using the strict consensus tree in Mesquite or TNT.
The justification for a new taxon is not strong at all. I have commented on the discussion in the text; please check it. However, the material resembles very similar to Shidaisaurus, and the features that are discussed in their overlapping material are not enough to justify that they are not the same taxon. Some of them could be preservation artifacts or need better justification or explanation. Please check the comments in the text.

Additional comments

The description of a new specimen of a metriacanthosaurid from China is necessary and relevant for the paleobiology of theropods. Most of the specimens of this relevant group of allosauroids are poorly described, non-accessible, poorly figured, and/or in Chinese. Therefore, it is so great to have new material, English-described, well-figured, and in an open-access journal, to make the knowledge of this fantastic group of dinosaurs accessible. Therefore, I would like to congratulate the authors of this study. I hope it will be published soon if they follow the major revision that could improve the manuscript. The most important fact is that naming this material as a new species is not justified enough, so the authors might see all the comments and questions about that in the text and include a better and stronger diagnosis of this new material as a new taxon. Finally, the phylogenetic methodology could be improved

---

## Round 0.2 · Minor Revisions

· Academic Editor

Minor Revisions

The manuscript is much improved from the previous version and I congratulate the authors for their meticulous work. However, I agree with Reviewer 2 that there are still a couple areas for revision, particularly expansion on the geological significance and specifically comparing the new taxon with other metriacanthosaurids. Once these matters are addressed, the manuscript should be ready for acceptance.

·

Basic reporting

The English has improved—nothing else to comment.

Experimental design

The phylogenetic analysis is now more complete. Although I do not entirely agree with not using ordered characters, especially for continuous ones as measurements, as the Carrano et al. 2012 methodology uses unordered characters, I understand the authors' decision. Nothing else to comment.

Validity of the findings

This has a significant impact due to the need for a good description of Metriacanthosaurids from China. Congrats again to the authors. The description has improved considerably.

Additional comments

After the revision, the manuscript has significantly improved. I think they follow most of the suggestions of both reviewers, and I think the manuscript should be accepted as is. Congratulations to the authors for their study!

·

Basic reporting

Language clarity and consistency
Overall, the writing language is acceptable. There are issues with word choice being unclear, incorrect, or not accurately conveying the authors intentions. For example, line 577: “The supraoccipital is seriously broken…” “Seriously” is not the appropriate word here. Better options would be: “largely incomplete”, “mostly broken”, or just succinctly stating that “The supraoccipital is broken and missing the dorsal part.” Various issues like this arise throughout the manuscript but are not so problematic as to warrant a complete revision. I’ve made specific comments below of some of the more problematic of these instances. I urge the authors to carefully review the manuscript for additional grammar, vocabulary, and other syntax errors.
Geologic Setting
I would like to see more information on the geological significance, specifically with reference to time, even if its just references to previous literature. One issue with a lot of new dinosaur discoveries is that the geology seems to get glossed over. Metriacanthosaurid papers specifically seem to be particularly bad in an overall lack of geologic data. However, the geology is important as it is our key window into the time component of evolution. Any future studies hoping to investigate tempo and mode of evolution for metriacanthosaurids will be crippled by a lack of chronostratigraphic data for the taxon. Adding a short section on geology shouldn’t be too difficult of an addition and will be of great importance to future work.
Descriptive elements
One of my personal pet peeves is stating what bones are preserved in the description for that bone itself. For example, Line 682-683: “Both the left and right dentaries are preserved, but their posterior boundaries are broken, so the suture with the surangular and angular is not clear.” In my opinion, a description is better when you focus solely on the morphology that you can see and forget the stuff that you can’t. In the above case, the fact that both dentaries are preserved is not directly useful to understanding the morphology of the new species. Nor is the fact that the contact surface isn’t observable (except in very specific cases where there is a very taxonomically/phylogenetically useful character that’s missing in that one area). The stuff that you can’t see acts only as filler and serves to only bog down the reader with useless information. A preservation section can be included before the general description that discusses what bones are present and how they are preserved. Again, personal preference, but it does make reading and gaining the important information of the description much easier.

Experimental design

Description: While I am by no means an expert on metriacanthosaurids, the descriptive and comparative aspect of the research are largely sound. Overall, this part of the investigation is well-defined, and some appropriate comparisons are made throughout the text. The only real issue I have with the description is that it often focuses on comparisons with other non-metriacanthosaurid and oftentimes even non-allosauroid taxa. I would like to see more specific comparisons of morphologies with other metriacanthosaurids. How is it similar or different to other taxa. This is done some throughout the descriptive text, so I am not saying it is absent, more so that I would like to see a greater comparison of ingroup taxa rather than broader comparisons with unrelated taxonomic groups. For example, the lacrimal and jugal of Yuanmouraptor are both quite different from that of Sinraptor but this isn’t really indicated by the description text.
Phylogenetic Analysis: I would like to see more done here. Overall, node support is somewhat low. Is this due to inclusion of taxa known from scrappy material? Does the MPT topology changed or is it strengthened by the exclusion of poorly represented taxa (e.g., taxa with low character scorability)? Would this improve the strength of the topology of Metriacanthosauridae? Another option that would be beneficial would be to perform a bootstrap analysis for node support as well as Templeton’s test for branch rearrangement and statistics.

Validity of the findings

The principal findings of the manuscript are largely valid. Yuanmouraptor does indeed appear represent a distinct new genus and species of Metriacanthosauridae. I think more could be done to bolster this claim as well overall strengthen of the cladistics analysis. Conclusion and interpretations are well stated and decently well argued.

Additional comments

355: “the quadratojugal ramus of THE jugal bifurcates…”
577: “Seriously broken”. Use “largely incomplete”, “mostly broken”, or just succinctly state that “The supraoccipital is broken and missing the dorsal part.”
589: “there are two foramina THAT PENETRATE the external surface…”
630: “The prootic contacts the laterosphenoid anteriorly, and posterior to the contact a shallow recess is presented” This implies that the prootic is anterior to the laterosphenoid, which is almost certainly not the case. The laterosphenoid is anterior to the prootic. The above are also two separate subjects and can be made into two sentences. Change to “THE PROOTIC IS SITUATED POSTERIOR TO THE LATEROSPHENOID. THE LATERAL SURFACE OF THE PROOTIC IS SHALLOWLY RECESSED.”
787: “All vertebrae are well preserved but none of them bears a rib.” This implies that you have a 100% complete vertebral series and also that Yuanmouraptor did not have ribs at all. Reword.
* * *
There are occasional issues throughout the text that are like the above. I recommend the authors thoroughly read through the manuscript to locate any grammar, vocabulary, and other syntax issues prior to re submission.

---

## Round 0.3 · accepted · Accept

· Academic Editor

Accept

The manuscript is much improved from the previous versions and the authors offered concrete and justified responses to reviewers' comments. Congratulations to the authors!

As a side note, before publication, double-check grammatical and visual aspects of the manuscript throughout (e.g., letters are much larger in Figure 15 than in Figure 16; lower-/uppercase use of Allosauroidea vs. allosauroids) to ensure that everything is consistent.